biomechanics/biotechnology/physiology

running, amputee, athletics, track, prostheses, 400 m

**Author for correspondence:**
Alena M. Grabowski
e-mail: alena.grabowski@colorado.edu

# Sprinting with prosthetic versus biological legs: insight from experimental data

Owen N. Beck[1], Paolo Taboga[2] and
Alena M. Grabowski[3,4]

[1]The Wallace H. Coulter Department of Biomedical Engineering, Georgia Institute of Technology and Emory University, Atlanta, GA, USA
[2]Department of Kinesiology, California State University, Sacramento, CA, USA
[3]Department of Integrative Physiology, University of Colorado, Boulder, CO, USA
[4]Department of Veterans Affairs, Eastern Colorado Healthcare System, Denver, CO, USA

ONB, 0000-0002-1802-6036; PT, 0000-0001-6529-8299;
AMG, 0000-0002-4432-618X

Running-prostheses have enabled exceptional athletes with bilateral leg amputations to surpass Olympic 400 m athletics qualifying standards. Due to the world-class performances and relatively fast race finishes of these athletes, many people assume that running-prostheses provide users an unfair advantage over biologically legged competitors during long sprint races. These assumptions have led athletics governing bodies to prohibit the use of running-prostheses in sanctioned non-amputee (NA) competitions, such as at the Olympics. However, here we show that no athlete with bilateral leg amputations using running-prostheses, including the fastest such athlete, exhibits a single 400 m running performance metric that is better than those achieved by NA athletes. Specifically, the best experimentally measured maximum running velocity and sprint endurance profile of athletes with prosthetic legs are similar to, but not better than those of NA athletes. Further, the best experimentally measured initial race acceleration (from 0 to 20 m), maximum velocity around curves, and velocity at aerobic capacity of athletes with prosthetic legs were 40%, 1–3% and 19% slower compared to NA athletes, respectively. Therefore, based on these 400 m performance metrics, use of prosthetic legs during 400 m running races is not unequivocally advantageous compared to the use of biological legs.

## 1. Introduction

Two male athletes with bilateral leg (transtibial) amputations have run 400 m faster than the Olympic athletics (track and field) qualifying standard. These athletes achieved world-class

performances by combining their unique physiology with passive-elastic carbon-fibre running-prostheses that act in-series with their residual limbs. Running-prostheses attach to residual limbs via rigid carbon-fibre sockets and emulate the spring-like behaviour of biological legs during ground contact [1,2]—but running-prostheses do not fully replicate biological leg function [1,3]. Unlike biological legs, running-prostheses cannot generate mechanical work de novo, neurally adjust geometry or change stiffness [4]. Despite functional differences, the use of running prostheses allows athletes with leg amputations to race 400 m shoulder-to-shoulder with non-amputee (NA) athletes at every competitive level—from youth athletics to the Olympic Games.

The potential for athletes with leg amputations to race alongside NA Olympians has been impeded by policymakers who have banned the use of running-prostheses from sanctioned NA competition [5]. These rules are founded on the assumption that the use of running prostheses provides an overall unfair advantage over the use of biological legs. Namely, the international governing body for the sport of athletics (World Athletics) enacted a rule from 2015 to 2020 that prohibited the use of a mechanical aid (e.g. running-prostheses) from sanctioned events unless an athlete could establish on the balance of probabilities that the use of such aid does not provide them an overall advantage over competitors not using such an aid [6]. In October 2020, the Court of Arbitration for Sport determined that this rule is 'discriminatory… unlawful, and invalid' and mandated World Athletics to bear the burden of proof regarding the exclusion of such mechanical aids [6]. Consequently, athletes with leg amputations are currently allowed to compete in sanctioned NA events unless World Athletics presents compelling evidence suggesting that the use of running prostheses provide users an unfair advantage over their competitors.

Athletics regulations regarding the use of running prostheses are hindered by the lack of scientific consensus regarding the *net effect* of running with prosthetic versus biological legs [7,8]; prosthetic legs include biological residual limbs, sockets and running prostheses. Currently, some scientists posit that using prosthetic versus biological legs enable athletes to achieve faster maximum running velocities [7] and run while expending less metabolic energy (better running economy) [9], factors that presumably improve running performance [10]. Alternatively, other scientists, including those from our research group, propose that the use of prosthetic versus biological legs slows an athlete's acceleration at the start of a race [11,12], as well as reduce maximum straightaway [3] and curve running velocity [13], factors that presumably worsen running performance. Not only is it difficult to weigh the importance of these purported 'pros and cons', but many hypothetical performance differences between athletes using prosthetic or biological legs have been contested by experimental data [14,15].

Rather than reiterating theoretical arguments [7,8], the goal of this study was to compare the data of athletes using bilateral prosthetic versus biological legs in experimental tests that relate to 400 m performance. To accomplish this goal, we measured the following 400 m performance metrics from the athlete who ran the fastest-ever 400 m time using prosthetic legs (fastest BA) following his competition season where he ran 400 m in 44.42 s: initial race acceleration [11,16], maximum straightaway running velocity [3,17–19], maximum curve running velocity [13,20,21], running velocity at aerobic capacity ($v\dot{V}o_{2peak}$) [17,22,23] and sprint endurance [10,17,24,25]. For context, a 44.42 s 400 m performance would have placed sixth at the 2021 Olympic Men's Finals. After testing the fastest BA's ability to perform each 400 m performance metric, we compared his results to those of other athletes with bilateral leg amputations using running-prostheses, including the second fastest such 400 m athlete in history (2nd fastest BA) [9,17]. Subsequently, we compared the best performance metric value achieved across all athletes with prosthetic legs to those across all NA athletes. If any athlete with prosthetic legs exhibited a 400 m performance metric that was better than that observed by the best NA athlete or over two standard deviations better than the average of elite NA athletes (consistent with [17]), prosthetic legs likely confer a specific advantage in that metric compared to biological legs.

# 2. Results: 400 m performance metrics

## 2.1. Initial acceleration

At the beginning of a 400 m race, athletes accelerate from a stationary starting-block position and around the track's initial curve. The average time that it took the fastest BA to sprint 20 m from a stationary starting-block position (Avg ± s.d.: 4.13 ± 0.10 s) was 40% slower (greater than 59 s.d.) than that of elite NA athletes who had 100 m personal records (PRs) that ranged from 9.95 to 10.29 s [16]. To our knowledge, no other athlete with bilateral prosthetic legs has had their 0 to 20 m running time published. Mechanically, the fastest BA's duration of force generation on the blocks was less than 1 s.d. different from those of elite NA athletes

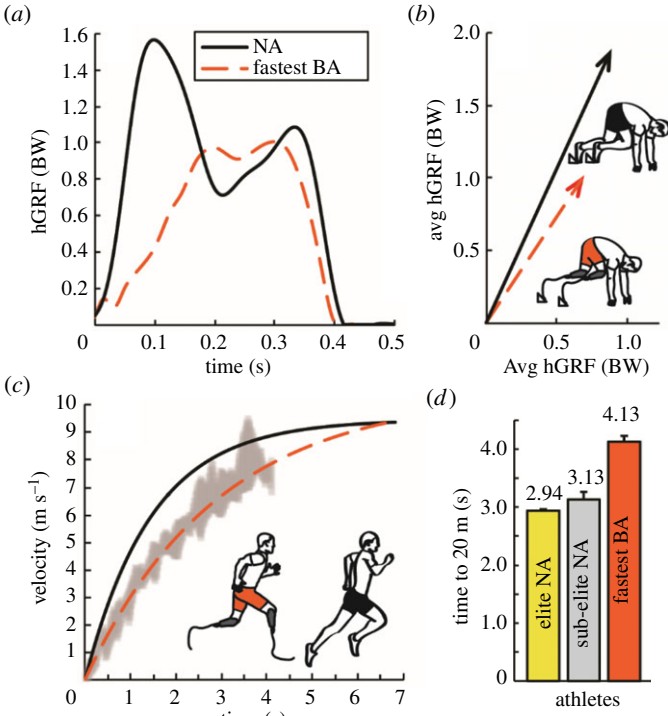

**Figure 1.** (*a*) The sum of the average (Avg) horizontal ground reaction force (hGRF) relative to body weight (BW) on the starting blocks versus time from the front and back legs of the fastest 400 m athlete with bilateral leg amputations using running-prostheses (fastest BA, red dashed line), and sub-elite non-amputee athletes (NA, black solid line) [11]. (*b*) Avg vertical ground reaction force (vGRF) versus hGRF on the starting blocks; arrows represent the average resultant GRF vector for the fastest BA (red dashed line) and NA athletes (black solid line) [26]. (*c*) Horizontal velocity (*v*) as a function of time (*t*) for the fastest BA (red dashed line) and for sub-elite NA athletes with 100 m personal records of 11.3 ± 0.35 s (black solid line: $v(t) = 9.46(1 - e^{-t/1.47})$) [27]. The red dashed line represents the average and grey area represents ± s.d. of the velocity versus time data collected from the fastest BA's three maximum acceleration trials. (*d*) The time it takes to accelerate from stationary starting blocks to 20 m for elite NA athletes (gold), sub-elite NA athletes (silver) [16] and the fastest BA (red). Error bars for NA athletes indicate SE across athletes and for the fastest BA indicates s.d. across three trials.

(0.362 versus 0.372 ± 0.13 s; Avg ± s.d.), thus the fastest BA's inferior 20 m performance versus elite NA athletes was related to his 31% lower mass-normalized horizontal force on the starting blocks (acceleration = force/mass) resulting in a 32% slower horizontal velocity exiting the starting blocks (figure 1 and table 1) [12]. Altogether, no experimentally tested athlete with prosthetic legs has accelerated out of the starting blocks and run faster than elite NA athletes over 20 m.

## 2.2. Maximum running velocity

After accelerating around the track's initial curve, 400 m athletes race along a straightaway. The athlete with the fastest maximum velocity can out-perform their competitors at a matched relative intensity and cover more distance for a given duration [10]. The fastest BA's maximum treadmill-running velocity was faster than that of any other athlete with bilateral prosthetic legs (11.4 m s$^{-1}$) [17,28], and similar to, but not faster than that of the fastest treadmill-tested NA athlete (11.72 m s$^{-1}$) [29] or athlete with a unilateral leg amputation (11.55 m s$^{-1}$) [18].

Both the fastest BA and elite male NA athletes run at approximately 10 m s$^{-1}$ on the straightaway from 100 to 200 m during 400 m races [30]. Thus, we compared the ground reaction force (GRF) parameters and step kinematics that govern running velocity for the fastest BA and NA athletes during treadmill running at 10 m s$^{-1}$. Briefly, running velocity (*v*) equals the product of stance average vertical GRF ($v$GRF$_{AVG}$) relative to body weight (BW), the horizontal distance travelled by the body's centre of mass (contact length; $L_c$), and step frequency (Freq$_{step}$) (see Methods for more detail) [19]:

$$v = \frac{v\text{GRF}_{\text{AVG}}}{\text{BW}} \cdot L_c \cdot \text{Freq}_{\text{step}}. \tag{2.1}$$

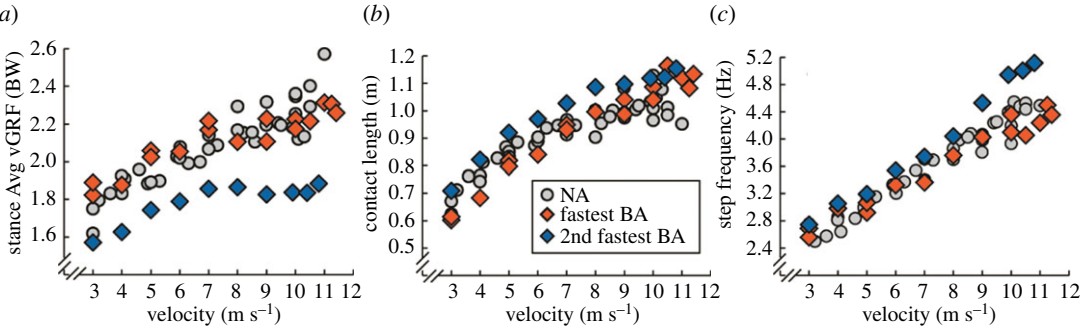

**Figure 2.** The fastest athlete with prosthetic legs uses similar biomechanics across running velocities as non-amputee 400 m athletes. (*a*) Stance average vertical ground reaction force (Avg vGRF) normalized to body weight (BW), (*b*) contact length and (*c*) step frequency versus velocity for non-amputee athletes (NA, silver circles) [17] and for the fastest (fastest BA, red diamonds) and second fastest (2nd fastest BA, blue diamonds) athletes with bilateral leg amputations using running-prostheses. At 10 m s$^{-1}$, the fastest BA generated 5% lower relative stance Avg vGRFs, 7% longer contact lengths and times and 1% faster step frequencies compared to non-amputee athletes (all parameters less than 2 s.d. from NA athlete Avg) [17].

**Table 1.** Starting acceleration biomechanics for athletes with prosthetic legs and non-amputee athletes. Average (Avg) vertical (vGRF) and horizontal (hGRF) ground reaction forces (GRFs) on the starting blocks, force application time on the starting blocks (time) and horizontal velocity out of the starting blocks for the fastest 400 m athlete with bilateral leg amputations using prosthetic legs (fastest BA), non-amputee athletes (NA), and athletes with unilateral leg amputations (UA). The average vGRF and hGRF for the virtual BA modelled in Taboga *et al.* [11] are calculated by averaging the forces from the prosthetic leg of athletes with UA. Notably, Mero *et al.* [26] report net vGRF values (i.e. net vGRF = total vGRF − body weight (BW)) and we report total vGRF to allow for comparisons with other reported values.

| source | athletes | 100 m PR (s) | Avg vGRF (BW) | Avg hGRF (BW) | time (s) | horizontal velocity (m s$^{-1}$) |
|---|---|---|---|---|---|---|
| current study | fastest BA | 10.91 | 1.01 | 0.68 | 0.362 | 2.44 |
| Taboga *et al.* [11] | virtual BA | n.a. | 1.00 | 0.60 | n.a. | n.a. |
| | recreational NA | 12.49 | 1.16 | 0.78 | 0.497 | 3.09 |
| | sub-elite UA | 13.17 | 1.16 | 0.72 | 0.417 | 2.80 |
| Mero *et al.* [26] | sub-elite NA | 10.80 | 1.87 | 0.89 | 0.361 | 3.22 |
| | non-elite NA | 11.50 | 1.64 | 0.70 | 0.368 | 2.94 |
| Rabita *et al.* [16] | sub-elite NA | 10.40–10.60 | n.a. | 0.79 | 0.412 | 3.17 |
| | elite NA | 9.95–10.29 | n.a. | 0.98 | 0.376 | 3.61 |

Overall, the fastest BA's stance average vertical GRF, contact length (and contact time) and step frequency, which includes aerial time and leg swing time, were similar to those of NA athletes (less than 8% and less than 2 s.d. from the Avg of NA athlete values) [17] (figures 2 and 3). Further, compared to the fastest BA, the 2nd fastest BA used stiffer running-prostheses [4,9], produced 19–23% lower stance average vertical GRFs relative to body weight, and took 14% shorter and more frequent steps, affirming that both athlete physiology and prosthetic configuration affect running biomechanics [28,31]. Therefore, the biomechanics that govern running velocity are not always similar within and across athletes with and without leg amputations (figures 2 and 3). Yet, using his current prosthetic configuration, the fastest BA achieves maximum running velocity using GRF parameters and step kinematics that are non-different from those of NA athletes.

## 2.3. Curve running

Athletes run along a curve for over half of a 400 m race, which is notable because athletes run slower on curves than on a straightaway [13,21]. On a counterclockwise curve with regulation outdoor track dimensions for lane 1 (radius: 36.5 m), the fastest BA's maximum over ground velocity was 6% slower than on a straightaway. To our knowledge, no other athlete with bilateral prosthetic legs has had their

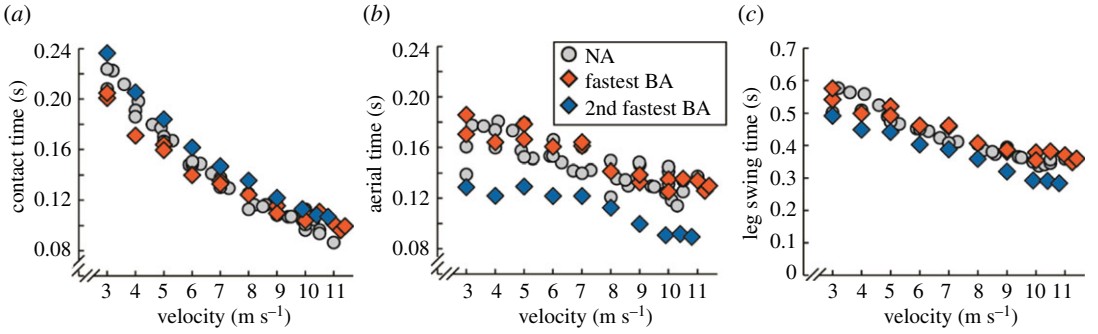

**Figure 3.** Biomechanics for the two fastest athletes with prosthetic legs and non-amputee athletes across running velocities. (a) Contact time, (b) aerial time and (c) leg swing time versus running velocity for non-amputee athletes (NA, silver circles), and for the fastest (fastest BA, red diamonds) and second fastest (2nd fastest BA, blue diamonds) 400 m athletes with bilateral leg amputations using running-prostheses [17].

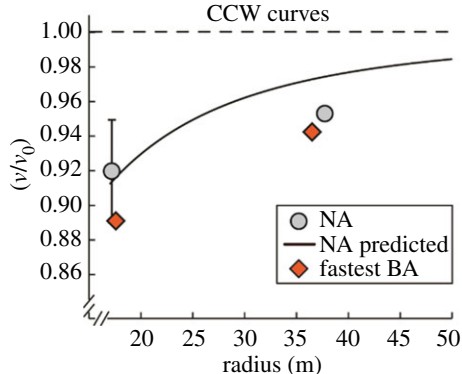

**Figure 4.** Maximum curve running velocities (v) relative to straightaway velocities ($v_0$) as a function of curve radius in the counterclockwise (CCW) direction. The dashed horizontal line represents the velocity for the straightaway running trials and the solid line is derived from equation (6.3) [21]. Maximum curve running velocity trials for non-amputee athletes (NA, silver circles), and the fastest 400 m athlete with bilateral leg amputations (fastest BA, red diamonds) using running-prostheses on different curve radii [21,32]. Error bars indicate s.d.

maximum curve and straightaway running velocities reported. For comparison, on the same and similar curve radius (36.5 m and 37.72 m), the maximum running velocity of NA athletes is reported to be 3% and 4.7% slower than on a straightaway, respectively (figure 4) [21,32]. Based on these data, the fastest BA does not have a relatively faster maximum curve running velocity than previously tested NA athletes.

## 2.4. Velocity at aerobic capacity

During a 400 m race, athletes expend metabolic energy via both anaerobic and aerobic metabolism [33]. If other performance metrics are equal, the athlete who has a faster velocity at aerobic capacity ($v\dot{V}o_{2peak}$) will out-perform others in a 400 m race [10,24]. The fastest BA's $v\dot{V}o_{2peak}$ (4.3 m s$^{-1}$) was 14% slower than that reported by the 2nd fastest BA ($v\dot{V}o_{2peak}$: 5.0 m s$^{-1}$) [17]. The $v\dot{V}o_{2peak}$ of the 2nd fastest BA is nearly identical to the average $v\dot{V}o_{2peak}$ of NA 400 m athletes (400 m PR: less than or equal to 48.03 s; Avg ± s.d.: 4.9 ± 0.04 m s$^{-1}$) [17] and 19% (greater than 3 s.d.) slower than that of NA distance runners with 10 km PRs under 32 min (figure 5) [34].

Because $v\dot{V}o_{2peak}$ depends on running economy and aerobic capacity ($\dot{V}o_{2peak}$), we also compared these parameters between athletes with and without bilateral leg amputations. The fastest BA's average running economy (160 ml O$_2$ kg$^{-1}$ km$^{-1}$ from 2.5 to 3.5 m s$^{-1}$) was better than any other athlete with prosthetic legs (table 2) [15]. This value was also 19% better (greater than 8 s.d.) than NA 400 m athletes (400 m PRs: 45.63 and 48.33 s) and non-different (less than 1 s.d.) from NA distance runners (5 km PRs: 13:34 to 13:59 m:s; 10 km PRs: 28:36 to 29:21 m:s) [37]. On the other hand, the fastest BA's $\dot{V}o_{2peak}$ (41.2 ml O$_2$ kg$^{-1}$ min$^{-1}$) was 22% lower than that of the 2nd fastest BA (52.7 ml O$_2$ kg$^{-1}$ min$^{-1}$) [17]. The $\dot{V}o_{2peak}$ of the 2nd fastest BA is 17% (greater than 2 s.d.) and 33% (greater than 6 s.d.) lower than that of the same NA 400 m athletes and NA distance runners [37], respectively. Thus, despite being relatively economical runners [15], the

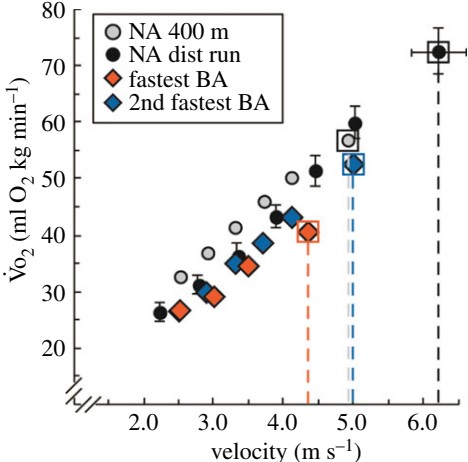

**Figure 5.** Submaximal and maximal rates of oxygen uptake ($\dot{V}o_2$) versus running velocity for non-amputee 400 m athletes (NA 400 m, silver circles) [17], non-amputee distance runners (NA dist run, black circles) [34] and the fastest (fastest BA, red diamonds) and second fastest (2nd fastest BA, blue diamonds) 400 m athletes with bilateral leg amputations using running-prostheses. The greatest $\dot{V}o_2$ value ($\dot{V}o_{2peak}$) for each athlete and cohort is indicated by a square around the symbol. The vertical dashed lines indicate the velocity at $\dot{V}o_{2peak}$. Error bars are s.d. when applicable.

**Table 2.** Aerobic metabolism and perceived exertion for the fastest athlete with prosthetic legs across a range of submaximal running velocities. Oxygen uptake, blood lactate concentration and Borg-scale rating of perceived exertion (RPE) [35] from the fastest 400 m athlete with bilateral leg amputations using running-prostheses (fastest BA) during the standing trial and constant-velocity running trials.

| velocity (m s$^{-1}$) | $\dot{V}o_2$ (ml $O_2$ kg$^{-1}$ min$^{-1}$) | blood lactate (mmol l$^{-1}$) | RER ($\dot{V}co_2/\dot{V}o_2$) | RPE (Borg) |
|---|---|---|---|---|
| 0 (standing) | 6.0 | n.a. | 0.83 | n.a. |
| 2.5 | 25.4 | 1.07 | 0.82 | 9–10 |
| 3.0 | 27.8 | 1.61 | 0.82 | 12 |
| 3.5 | 32.9 | 2.48 | 0.89 | 14 |
| 4.0[a] | 38.0[a] | 5.48[a] | 1.01[a] | 19[a] |

[a]We did not include data from 4.0 m/s in our analyses because the fastest BA's blood lactate measurements were greater than 4 mmol l$^{-1}$ and respiratory exchange ratio (RER) was greater than 1.00 [36].

lower $\dot{V}o_{2peak}$ of the measured athletes with prosthetic legs contribute to a v$\dot{V}o_{2peak}$ that is not faster than that of NA 400 m athletes and distance runners.

## 2.5. Sprint endurance

The fastest running velocities that NA athletes can maintain for approximately 10 to 300 s are remarkably well predicted by a simple model that incorporates their maximum running velocity and v$\dot{V}o_{2peak}$ [10,24]. The fastest BA performed six all-out treadmill-running trials at different velocities that a model derived from NA data predicted he could maximally sustain for 14 to 133 s. Similar to the results of the only other such athlete to complete this protocol (2nd fastest BA) [17], the duration that the fastest BA could maintain each relative running velocity was nearly identical to that predicted from the NA model (less than 3% and less than 1 s.d.) (figure 6 and table 3). Hence, despite their relatively fast race finishes [30], the studied 400 m athletes with prosthetic legs do not appear to have better sprint endurance profiles compared to NA athletes.

# 3. 400 m race splits

In addition to the 400 m performance metrics, we compared 400 m race splits for the fastest BA and elite male NA athletes. We calculated the fastest BA's 100 m splits from his fastest 400 m race prior to participating in this study and compared them to those of elite male NA athletes during the 2017

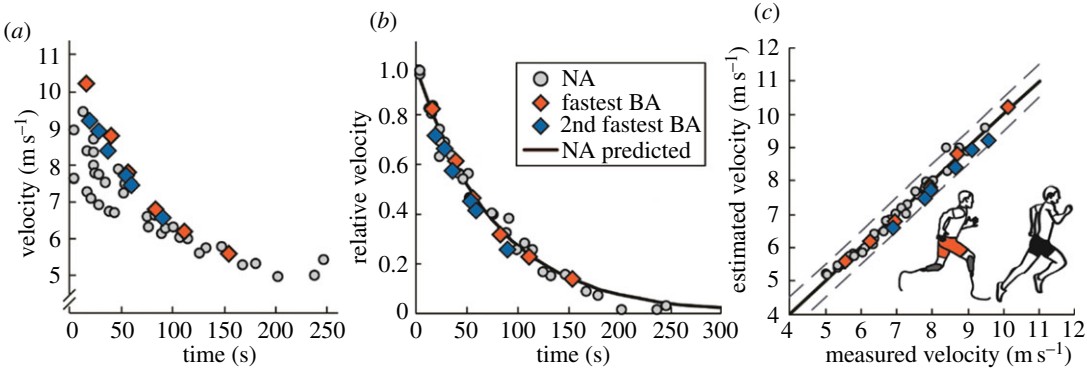

**Figure 6.** Sprint endurance profiles for non-amputee athletes (NA, silver circles) [17], and from the fastest (fastest BA, red diamonds) and second fastest (2nd fastest BA, blue diamonds) 400 m athletes with bilateral leg amputations using running-prostheses [17]. (*a*) Time that athletes can sustain a given velocity. (*b*) Time that athletes can sustain a relative velocity, as well as a model-fit from NA data (black line) [10,17,25]. Relative velocity equals $(v(t) - v\dot{V}_{O_{2peak}})/(v_{max} - v\dot{V}_{O_{2peak}})$, where $v(t)$ is measured velocity for a given time ($t$), $v\dot{V}_{O_{2peak}}$ is velocity at aerobic capacity and $v_{max}$ is maximum velocity. (*c*) Estimated versus measured velocity for each time. Black line indicates the line of identity and dashed lines indicate the SE from NA athletes [17].

**Table 3.** The sprint endurance time for a given velocity is nearly identical for the fastest athlete with prosthetic legs (fastest BA) and non-amputee (NA) athletes. Sprint endurance velocity that the fastest 400 m athlete with bilateral leg amputations using running-prostheses could sustain for a given time, and the corresponding velocity modelled for non-amputee athletes with the same maximum running velocity and velocity at aerobic capacity (equation (6.9)) [10].

| fastest BA time (s) | fastest BA velocity (m s$^{-1}$) | NA velocity (m s$^{-1}$) |
| --- | --- | --- |
| 16.2 | 10.2 | 10.1 |
| 39.7 | 8.8 | 8.6 |
| 55.9 | 7.8 | 7.8 |
| 83.1 | 6.8 | 6.7 |
| 111.2 | 6.2 | 6.0 |
| 154.4 | 5.6 | 5.3 |

International Association of Athletics Federations (IAAF) World Championships 400 m final [30]. Overall, the fastest BA's 400 m time (44.42 s) was less than 1 s.d. from the average and within the range of elite NA athlete 400 m times (table 4) [30]. During the initial 100 m, the fastest BA was 8.3% slower (greater than 7 s.d.) than the elite NA athletes [30]. Over the second and third 100 m sections, the fastest BA was 2.5% slower (less than 2 s.d.) and 0.2% faster (less than 2 s.d.) compared to the elite NA athletes, respectively [30]. Over the fourth and final 100 m, the fastest BA ran 9.9% faster (greater than 3 s.d.) than the elite NA athletes (table 4) [30]. Thus, based on a 2 s.d. cut-off, the fastest BA's four 100 m race splits were slower, non-different, non-different, and faster than those of elite NA athletes from the 2017 IAAF World Championships, resulting in similar 400 m race times between the fastest BA and elite NA athletes (less than 1 s.d.; table 4).

How do the fastest BA's experimentally derived performance metrics compare to his 400 m race splits? *Acceleration.* From 0 to 100 m of his fastest 400 m race prior to this study, the fastest BA ran 4.3% faster than what we predicted based on his radar-gun data (figure 1) and equations (6.1) and (6.2) (see Methods). Notably, our experimental predictions do not include athlete reaction time, whereas the 0–100 m race split does (greater than 0.1 s). Further, the 4.3% difference in race time versus predicted time is comparable to that of NA athletes, who were 6.2% faster from 0 to 100 m of the 2017 IAAF World Championship final 400 m race compared to their predicted times [27]. *Maximum velocity.* During the fastest BA's fastest 400 m race split (from 100 to 200 m), he was 1.71 m s$^{-1}$ slower than his maximum treadmill-running velocity. For context, the fastest NA athlete race split was 1.63 m s$^{-1}$ slower than the maximum NA treadmill-running velocity [29]. There are many potential reasons for these experimental- versus race-based discrepancies. First, the fastest BA's

**Table 4.** Elite 400 m race splits for the fastest athlete with prosthetic legs (fastest BA) and elite male non-amputee athletes (NA). Running lane, consecutive 100 m race split times and 400 m race times for elite non-amputee athletes competing in the 400 m final of the 2017 International Association of Athletics Federations (IAAF) World Championship [30] and the fastest BA competing in Prague in 2018, where he ran his best time prior to this study.

| athlete | lane | time (s) | | | | |
| | | 0–100 m[a] | 100–200 m | 200–300 m | 300–400 m | 0–400 m |
| --- | --- | --- | --- | --- | --- | --- |
| Van Niekerk | 6 | 10.85 | 9.93 | 10.86 | 12.34 | 43.98 |
| Gardiner | 4 | 11.04 | 9.97 | 10.93 | 12.47 | 44.41 |
| *fastest BA* | *6* | *11.92* | *10.32* | *10.84* | *11.32* | *44.42* |
| Haroun | 3 | 11.26 | 10.33 | 11.06 | 11.83 | 44.48 |
| Thebe | 9 | 10.94 | 10.10 | 10.83 | 12.79 | 44.66 |
| Allen | 5 | 10.94 | 10.00 | 11.02 | 12.92 | 44.88 |
| Gaye | 8 | 11.01 | 9.98 | 11.28 | 12.77 | 45.04 |
| Kerley | 2 | 11.04 | 10.15 | 11.17 | 12.87 | 45.23 |
| NA Avg | 5.3 | 11.01 | 10.07 | 11.02 | 12.57 | 44.67 |
| s.d. | 2.6 | 0.13 | 0.14 | 0.16 | 0.39 | 0.42 |

[a]0–100 m split times include reaction time.

fastest race split (100–200 m) may have been slower than his maximum treadmill velocity because he ran further on the track versus the treadmill (31 versus 100 m) [10,25]. Second, athletes can run faster on a treadmill compared to overground because they do not need to overcome as much air resistance [38]. Third, athletes pace themselves during 400 m races but not during maximum running velocity treadmill trials [39]. Fourth, differences in the precision of calculating an athlete's running velocity using different methods (e.g. calibrated treadmill velocity versus 25 Hz video recording of 100 m race splits) could have potentially affected the experimental- versus race-based comparisons. *Curve running.* Over the third 100 m split, both the fastest BA and elite NA athletes slowed more than predicted (4.8% and 8.6%, respectively) based on their maximum curve running velocities. Athletes may have run relatively slower during their third race split than predicted due to racing a longer distance on the track versus the treadmill (20 versus 100 m), in addition to altered pacing strategies and fatigue, which are not present in the experiments that measured maximum curve running velocity. Further complicating the experimental- versus race-based comparison, the fastest BA and NA athletes [32] may have been accelerating throughout their straightaway and curve running experimental trials, whereas they were decelerating throughout the third race split (200–300 m) (table 4). *Sprint endurance.* Over the final 100 m split, the fastest BA was faster than elite NA athletes despite having a similar sprint endurance profile [7,10,24]. The difference between the sprint endurance profile versus the corresponding race splits may be related to typical variability in sprint endurance profiles, differences in race strategies and/or environmental conditions. Alternatively, prosthetic legs may enable athletes to sustain relatively fast velocities for a longer duration than biological legs, despite nearly identical experimentally derived sprint endurance profiles (figure 6).

# 4. Limitations and future directions

We acknowledge that it is uncertain exactly how fast an athlete with prosthetic legs could run 400 m if they were a NA athlete with biological legs using footwear (or vice versa). Additionally, there is currently no published model that accurately predicts 400 m performance. Thus, in this study, we compared 400 m performance metrics from athletes with bilateral leg amputations to those of NA athletes who were tested experimentally using similar protocols. Notably, the athlete comparisons were not exhaustive, were potentially statistically underpowered, and subtle differences between experiments may have influenced these comparisons (e.g. indoor versus outdoor track testing). We implore future studies to improve models of running performance and to use consistent protocols to compare data between studies. Further, more research is warranted to determine why the 400 m race splits of athletes with

bilateral leg amputations differ from those of NA athletes. Such future research will help reveal how biomechanical and physiological factors affect running performance, which can be used to inform athletics rules and regulations.

# 5. Conclusion

Currently, no athlete with bilateral leg amputations using passive-elastic carbon-fibre running-prostheses, including the fastest such athletes, has ever been reported to have a single 400 m performance metric that is better than that achieved by NA athletes. Therefore, based on experimentally derived 400 m performance metrics, athletes with bilateral leg amputations using passive running prostheses cannot be unequivocally considered to have an advantage over NA athletes during 400 m competitions.

# 6. Methods

## 6.1. Participant

The fastest 400 m athlete with bilateral leg (transtibial) amputations using running prostheses (fastest BA; age: 29 years; standing height with running-prostheses: 1.89 m; standing leg length (greater trochanter to ground): 1.07 m; mass without running-prostheses: 65.9 kg; mass of both running-prostheses: 2.5 kg) performed a series of tests over 5 days following his competition season when he ran a season-best 400 m in 44.42 s. For each test, the fastest BA used his competition passive-elastic carbon-fibre running-prostheses: Ottobock 1E90 Sprinter, stiffness category 3. The University of Colorado Boulder Biomedical Institutional Review Board (no. IRB00000774) approved the protocol. The participating athlete provided informed consent in accordance with the approved protocol prior to testing (Protocol: 18-0456).

## 6.2. Acceleration

During one of the testing days, the fastest BA warmed-up and then performed three maximum effort accelerations out of the starting blocks along a straightaway on an indoor track. The fastest BA placed the starting blocks in his typical competition position—each block was on top of a separate Mondo-covered (Mondo S.p.A., Italy) force plate (AMTI, Watertown, MA). We instructed the fastest BA to run as fast as possible through 20 m after hearing the starting commands. During each of these trials, we measured GRFs at 1000 Hz and horizontal velocity using a radar gun (Stalker ATS II radar system, Applied Concepts Inc., Richardson, Texas, USA) at 47 Hz. The radar gun was positioned 5 m behind the starting line and 1 m above the ground [27]. Between each acceleration trial, the fastest BA recovered for at least 5 min. We compared data from the fastest BA to those of NA athletes from a previous study who performed two maximum effort accelerations out of the starting blocks along a straightaway over distances of 0–10, 0–15, 0–20, 0–30 and 0–40 m [16].

We used a MATLAB script (MathWorks, Natick, MA) to determine the resultant GRFs that the fastest BA exerted on the starting blocks. We filtered the GRF data using a fourth-order Butterworth low-pass filter with a 30 Hz cut-off and identified the beginning and ending of the push-off phase as the instant when the total horizontal GRF crossed 20 and 1 N for the front and back block, respectively [11]. We used a higher horizontal GRF threshold (20 N) for the beginning of the push-off phase compared to Taboga et al. [11] (0 N) because the fastest BA's hands were not completely placed on the front force plate in the 'set' position. Without accounting for the fastest BA's reaction time, we recorded how long it took for him to run from 0 to 20 m ($t_{20\,m}$), and determined the corresponding velocity–time profile using the following model, which is consistent with previous studies [27,40]:

$$v(t) = v_{max}(1 - e^{-t/\tau}), \tag{6.1}$$

where $v(t)$ is the measured velocity as a function of time ($t$), $v_{max}$ is the athlete's calculated maximum velocity, e is the base of the natural logarithm and $\tau$ is a calculated time constant. We used MATLAB's Curve Fitting Toolbox to calculate $v_{max}$ and $\tau$ from the radar-gun data. Sequentially, we used $\tau$ to calculate maximum acceleration ($a_{max}$):

$$a_{max} = \frac{v_{max}}{\tau}. \tag{6.2}$$

We calculated $v_{\mathrm{max}}$, $\tau$ and $a_{\mathrm{max}}$ for each trial and averaged them to compare the fastest BA's biomechanics to those of NA athletes maximally accelerating from 0 to 20 m [27].

## 6.3. Maximum running velocity

On two separate days, the fastest BA warmed-up and then performed a series of constant-velocity running trials on a force-measuring treadmill (Treadmetrix, Park City, UT). We calibrated the treadmill speeds prior to running trials using a Shimpo Tachometer (Electromatic Equip't Co., Inc, Cedarhurst, NY). The fastest BA began each series of running trials at 3 m s$^{-1}$ and following each successful trial we incremented treadmill velocity 1 m s$^{-1}$ for the subsequent trial. A successful trial indicated that the fastest BA was able to maintain his anterior–posterior position on the treadmill while taking at least 12 consecutive steps [29,41]. As the fastest BA approached his maximum velocity, we implemented smaller treadmill velocity increments (e.g. +0.5 m s$^{-1}$). If the fastest BA was unable to maintain anterior–posterior position on the treadmill for at least 12 consecutive steps, the trial was considered unsuccessful, and he had the option to repeat the previous trial's velocity or deem the last successful trial his maximum velocity. The maximum velocity testing protocol was identical between the fastest BA, fastest athlete with a unilateral leg amputation [18] and fastest NA athletes [29]. The fastest BA had ad libitum rest between each trial. We measured GRFs throughout the duration of each trial at 1000 Hz, filtered them using a fourth-order low-pass Butterworth filter with a 30 Hz cut-off, and used the filtered data from 12 to 20 consecutive steps to calculate average GRF parameters and stride kinematics from equation (6.6) using a MATLAB script. We used a 20 N vertical GRF threshold to detect periods of ground contact.

Running velocity ($v$) is the product of stride length ($L_{\mathrm{stride}}$) and stride frequency ($\mathrm{Freq}_{\mathrm{stride}}$):

$$v = L_{\mathrm{stride}} \cdot \mathrm{Freq}_{\mathrm{stride}} \; . \tag{6.3}$$

Two steps comprise a stride, and steps are lengthened by producing greater stance average vertical GRF ($v\mathrm{GRF}_{\mathrm{AVG}}$) relative to body weight (BW) and/or increasing the horizontal distance travelled by the athlete's centre of mass during ground contact (contact length: $L_c$) [19,29].

$$L_{\mathrm{step}} = \frac{v\mathrm{GRF}_{\mathrm{AVG}}}{\mathrm{BW}} \cdot L_c \; . \tag{6.4}$$

We calculated step frequency ($\mathrm{Freq}_{\mathrm{step}}$) as the reciprocal of the sum of the ground contact time ($t_c$) and subsequent aerial time ($t_a$) [19,29]:

$$\mathrm{Freq}_{\mathrm{step}} = \frac{1}{(t_c + t_a)} \; . \tag{6.5}$$

Thus, running velocity equals the product of stance average vertical GRF relative to body weight, contact length and step frequency [19,29]:

$$v = \frac{v\mathrm{GRF}_{\mathrm{AVG}}}{\mathrm{BW}} \cdot L_c \cdot \frac{1}{(t_c + t_a)} . \tag{6.6}$$

## 6.4. Curve running

On a separate day, the fastest BA warmed-up and then performed maximum effort 40 m sprints on an outdoor track beginning from a standing start on a straightaway and counterclockwise curves that replicated lane 1 of a regulation 400 m outdoor track (radius; $r = 36.5$ m) and 200 m indoor track ($r = 17.2$ m) (figure 4) [13]. The fastest BA performed three sprints per straightaway and curve-radius condition (two curve radii), with at least 8 min of rest between each trial, which is consistent with the protocol performed by Taboga et al. [13]. Additionally, data from NA athletes [32] (grey symbols in figure 4) involved three 60 m sprints with 8 min of rest between each trial on two separate days. We recorded the sagittal plane view of each trial with a high-speed video camera (Casio EX -ZR1000, Casio Computer Co. Ltd, Japan) at 240 Hz, which we placed 50 m away from the straightaway and at the centre of each curve to minimize parallax (see electronic supplementary material) [13]. We determined each trial's running velocity between 20 and 40 m and normalized velocity for each curve running trial ($v$) to the velocity from the straightaway trial ($v_0$):

$$\frac{v}{v_0} \; . \tag{6.7}$$

Using Greene's model [21], we estimated the maximum curve running velocity of NA athletes:

$$\left(\frac{rg}{v_0^2}\right) = \frac{(v/v_0)^3}{\sqrt{1-(v/v_0)^2}},$$

(6.8)

where $r$ is the radius of the curve and $g$ is gravitational acceleration (9.81 m s$^{-2}$). We used a MATLAB script to numerically solve and plot $v/v_0$ at different radii for NA athletes and calculate the normalized curve running velocities for the fastest BA (figure 4).

## 6.5. Velocity at aerobic capacity

On a different day, the fastest BA arrived at the laboratory at least three hours postprandial. Upon arrival, he performed a 5-min standing trial while we measured his rates of oxygen consumption ($\dot{V}o_2$) and carbon dioxide production ($\dot{V}co_2$) using expired gas analysis (ParvoMedics TrueOne 2400, Sandy, UT, USA). Following a warm-up, he performed 5-min running trials at 2.5, 3.0, 3.5 and 4.0 m s$^{-1}$ on a treadmill (Treadmetrix, Park City, UT). Immediately following each running trial, the fastest BA briefly stood in place while we obtained approximately 50 µl of blood by pricking his finger to determine his blood lactate concentration (table 2). We monitored blood lactate concentration to ensure that the fastest BA primarily relied on aerobic metabolism during submaximal running trials, defined as a blood lactate level below 4 mmol l$^{-1}$ [22,42] and a respiratory exchange ratio (RER) less than 1.0. We analysed blood samples in duplicate with an YSI 2300 lactate analyser (YSI Inc., OH, USA) (table 2). After each blood sample, the fastest BA immediately initiated the subsequent running trial. After completing these four trials, the fastest BA rested for 10 min and then performed an aerobic capacity test. The aerobic capacity test began at 3.5 m s$^{-1}$ on a level treadmill and following each minute, we increased running velocity by 0.5 m s$^{-1}$ until the fastest BA reached exhaustion and terminated the test. Our protocol was similar to the protocols from comparison studies, which assessed steady-state rates of oxygen uptake during 4–7 min submaximal running trials [17,34,37], assessed blood lactate measures at the end of submaximal running trials [34,37], and performed incremental aerobic capacity tests 10–20 min after the last submaximal running trial [34,37].

We measured the fastest BA's $\dot{V}o_2$ and $\dot{V}co_2$ throughout each trial. We averaged $\dot{V}o_2$ and $\dot{V}co_2$ during the last 2 min of the standing trial and running trials, used the average $\dot{V}o_2$ to calculate steady-state rates of oxygen uptake, and used the ratio of $\dot{V}co_2$ and $\dot{V}o_2$ to calculate RER. We averaged $\dot{V}o_2$ over the final 15 s of the aerobic capacity test to determine the fastest BA's peak rate of oxygen uptake ($\dot{V}o_{2peak}$) [43]. We normalized the fastest BA's running economy and aerobic capacity using his total mass including his running-prostheses. Then, we determined the fastest BA's velocity at $\dot{V}o_{2peak}$ by linearly extrapolating his $\dot{V}o_2$ values versus running velocity from 2.5 to 3.5 m s$^{-1}$ [44].

## 6.6. Sprint endurance

On a separate day, the fastest BA performed six treadmill-running trials at velocities between his $v\dot{V}o_{2peak}$ and maximum velocity (5.6, 6.2, 6.8, 7.8, 8.8 and 10.2 m s$^{-1}$). We randomized the trial order. Each trial was initiated by the fastest BA lowering himself from the handrails onto the moving treadmill belt. We measured the time that the fastest BA could sustain each treadmill velocity using a stopwatch. We compared these data to those of previous studies that involved NA athletes and the 2nd fastest BA, who performed 2–6 sprint endurance trials per session, totaling 6–15 trials per participant [10,17].

Previous studies have demonstrated that heterogeneous NA athletes and the 2nd fastest BA can all sustain the same running velocities normalized to their $v\dot{V}o_{2peak}$ and maximum velocity ($v_{max}$) for the same amount of time [10,17,25]. Specifically, the velocity ($v$) that can be sustained for a time ($t$) by any athlete is well predicted ($R^2 = 0.94$) from $v_{max}$, $v\dot{V}o_{2peak}$, according to equation (6.9) [10]:

$$v(t) = v\dot{V}_{O_2peak} + (v_{max} - v\dot{V}_{O_2peak})e^{-kt},$$

(6.9)

where e is the base of the natural logarithm and $k$ is an exponential constant for running that describes the decrement in velocity that occurs at progressively longer times (up to approx. 300 s) [10,25]. To compare the fastest BA to NA athletes, we calculated the velocities that could be sustained by an athlete with the same $v_{max}$ and $v\dot{V}o_{2peak}$ as the fastest BA and an exponential constant that was previously validated from NA athletes ($k = 0.013$) [10].

## 6.7. 400 m race splits

We quantified 100 m splits from the fastest BA and elite male NA athletes during outdoor 400 m races. Specifically, we determined the fastest BA's 100 m splits from a video recording of a 400 m race where he ran 44.42 s (https://youtu.be/rqIvKYOlltw) using digitizing software (Kinovea). From a published report [30], we indexed 100 m race splits of elite male NA sprinters during the 400 m final of the IAAF World Championships in London, UK.

## 6.8. Statistics

We deemed the 400 m performance metrics of athletes with prosthetic legs to be different from those of NA athletes when their value fell outside the range observed by NA athletes or when our statistical tests revealed greater than or equal to 95% confidence that they were different from the comparison NA cohort ($p < 0.05$) (same as [17]). We assumed that NA data were normally distributed. Because less than 5% of normally distributed data fall outside two standard deviations from the average, we used two standard deviations from the average as our conservative statistical threshold. In other words, if any athlete with prosthetic legs exhibited a performance metric that was more than two standard deviations from the mean of a NA cohort, we would consider that athlete to exhibit a different performance metric from the corresponding NA cohort. We did not performance-match athletes with prosthetic versus biological legs because doing so would only reveal how each athlete achieves a given 400 m performance metric, not whether a given athlete performs better or worse. Additionally, because we quantified 400 m performance metrics from the fastest BA using protocols that emulated those of previous studies, our statistical comparisons were selective, not comprehensive, and potentially underpowered.

Ethics. The University of Colorado Boulder Biomedical Institutional Review Board (no. IRB00000774) approved the protocol. The participating athlete provided informed consent in accordance with the approved protocol prior to testing (Protocol: 18-0456).
Data accessibility. Experimental data from this study are located in the paper and electronic supplementary material.
Additional data are provided in the electronic supplementary material [45].
Authors' contributions. O.N.B.: conceptualization, data curation, formal analysis, investigation, methodology, validation, visualization, writing—original draft, writing—review and editing; P.T.: conceptualization, data curation, formal analysis, investigation, methodology, validation, visualization, writing—review and editing; A.M.G.: conceptualization, data curation, formal analysis, investigation, methodology, project administration, resources, software, supervision, validation, visualization, writing—review and editing.
All authors gave final approval for publication and agreed to be held accountable for the work performed therein.
Competing interests. The fastest BA provided funds to cover travel to and accommodations at the investigation site for two of the paper's authors (O.N.B. and P.T.). All authors provided their time pro bono throughout the entirety of this study, as well as during all legal cases. All authors were involved in the following legal cases: Court of Arbitration in Sport, Leeper versus World Athletics, 2020 and 2021.
Funding. We received no funding for this study.
Acknowledgements. We thank the fastest BA for participating in this study. We also thank members of the University of Colorado Boulder's Applied Biomechanics, Locomotion and Applied Exercise Science Laboratories for assisting with data collection. We thank Dr Rodger Kram and Dr Hugh Herr for suggestions on the manuscript.

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
