## [Peer Review File · Royal Society Open Science]

Review History

RSOS-211046.R0 (Original submission)

Review form: Reviewer 1

Is the manuscript scientifically sound in its present form?

Yes

Are the interpretations and conclusions justified by the results?

No

Is the language acceptable?

Yes

Do you have any ethical concerns with this paper?

No

Have you any concerns about statistical analyses in this paper?

Yes

Recommendation?

Major revision is needed (please make suggestions in comments)

Comments to the Author(s)

Dear authors,

This is a contemporary topic, and it will remain as such for years to come. Original information/data collected by established scientists and laboratories should be made available to the rest of the scientific community to study it further and debate it if necessary. Therefore, given the nature of the topic, my approach is to facilitate the dissemination of such research as it will enable us to synthesise a consensus on the use of artificial limbs in sport and make a positive difference to amputee and non-amputee athletes.

Whilst I am comfortable with your approach and the reasoning you are developing, I feel that that the key messages (abstract, conclusions and in other parts) are going beyond the evidence collected.

You have quantified for first time performance and physiological parameters from an elite amputee, and this is important for our knowledge and understanding but I suggest being cautious when trying to generalise the applicability of the findings. I would suggest stating that “the use of prosthetic legs during 400 m running races cannot be considered a priori advantageous” but I would refrain from more general statements referring to all amputee and non-amputee athletes and the application of your findings to other events. Apart from the small sample size the study is affected by a few limitations preventing us to draw robust and universal conclusions. For example, data have been collected in a different way than previous studies but also you are contrasting your athlete’s data against only a small range of previous studies which measured similar variables in non-amputee athletes. There are a lot more studies in the literature with data from sprint starts, sprinting mechanics and performance which you have not included in your comparative past literature. On some occasions you are using review articles (e.g., Mero et al., 1992) rather than the actual original research to pull relevant data. There are two options here: you either do a full search of literature and enrich your sources (and redo your statistics) or you keep the existing sources but make a clear statement that this is indicative literature. This has an implication in your statistical analysis. I recognise that the approach to use 2SDs as a boundary to identify similarities/dissimilarities has been used before in a similar setting/article but given your sources do not include all key previous studies/data this method is rather underpowered here. I would just present the whole project as a case study and provide comparisons only through absolute and relative differences. If you prefer to keep the 2SD metric too, please ensure that the limitations of this technique are discussed briefly. Furthermore, I would consider an addition to the title: Sprinting with prosthetic versus biological legs: an unfair advantage? Interpretations from authentic data (or something similar). Please consider the above and modify your introduction, core arguments, conclusions and abstract accordingly. A limitations section/paragraph should be included before drawing final conclusions.

Please see below more specific comments:

INTRODUCTION

P3, Line 6: replace “exceptional” with “male”. Their standing is demonstrated from information further down

P3, Line 32: change to “non-amputee Olympians”

P3, Line 32: replace “stifled” with “prevented” or similar

P3, Line 39: change from “the governing body of Olympic athletics” to “international governing body for the sport of athletics”

P4, Line 25: Change from “Alternatively, other scientists propose...” to “Alternatively, other scientists, including our research group, propose”

P4, Line 49: “Fastest athlete”? Need some more context, e.g., fastest on earth, fastest last year, or similar.

P4, Line 49: Add “following” between “the” and “400 m”

P5, line 7: Change “would have placed 7th at the 2016 Olympics” to “would have placed him 7th amongst the 2016 Olympic non-amputee finalists” or something similar.

METHODS

P19, Line 22: correct to “consent”

P20, Lines 8-13: What do you mean by “acceleration phase”? Is this the push-off phase? Please use a term more relevant to sprint start mechanics. Having a look at your 2014 sprint start paper (<http://dx.doi.org/10.1123/jab.2013-0113>) I could see different thresholds defined there. This is not a problem, but if there are different you may need a very brief justification, given you are citing this reference.

P22, Lines 8-10: Can you please clarify if these were three sprints per radius condition?

P22, Line 15: provide resolution information for the Casio camera. Also, report the zoom factor used as the camera was placed far away from the plane of action.

P23, Line 3: Specify make and model of treadmill

P23, Line 31: Specify make and model of gas analysis system

RESULTS

P5, Lines 43-45: By stating “No other athlete with prosthetic legs has had their lab-based 0 to 20 m running time published” do you imply that your test a was lab-test? If this is the intention, then you would need to remove the term “lab-based” as these were not pure lab conditions. Perhaps you can say, “experimentally recorded” or something similar.

P6, Lines 3-6: Please remove “Altogether, athletes with prosthetic legs do not accelerate faster than non-amputee athletes at the beginning of a race”.

Also remove the following from Figure 1 caption: “Athletes with prosthetic legs accelerate slower at the beginning of a race than non-amputee athletes”.

The above statements are too universal and are not supported by your data. Even though it appears plausible you cannot categorically say this as you have not experimentally tested all amputees and non-amputee athletes.

P7, Table 1: The data from the Mero et al (1992) review paper for the vGRF refers to the maximal and not average force from those studies (check again table I from Mero et al 1992).

P7, Line 55: How did you ensure that your measurements were taken at 10 m/s? Also, are you sure that the reference number 19 applies here, and not number 17? I can’t find any non-amputee data at 10m/s in Weyand et al, 2000.

P9, Lines 38-45: Please use this version: “Based on these data, the studied athletes with prosthetic legs do not have a relatively faster maximum curve running velocity than non-amputee athletes tested previously since their maximum curve-running velocity is ~1-3% slower than those of non-amputee athletes”.

P10, Line 39: Start the sentence as: “The velocity of 5.0 m/s ...”.

P11, Figure 5 caption: Remove “Non-amputee athletes have a faster running velocity at aerobic capacity than athletes with prosthetic legs”.

P11, Line 52: replace “worse” with “lower”

P12, Line 45: use “the studied 400 m athletes...”

P14, Line 29: remove “statistically”

P15, Table 4: how did you obtain the split times for the Fastest BA? Was it from TV footage? Please provide relevant information in the methodology section.

P15, main paragraph: As suggested earlier, please refrain from using lab-based for track measurements with the radar gun and replace with “experimental” or similar.

P15, Lines 53 – P16, Line 10: I would add one more reason for the discrepancies and this is the fact that the velocities have been calculated via different methods (split times vs treadmill mechanics)
BL400 Data file: Starts Tab: Peak hGRF is not in BW as indicated.

Review form: Reviewer 2

Is the manuscript scientifically sound in its present form?

Yes

Are the interpretations and conclusions justified by the results?

Yes

Is the language acceptable?

Yes

Do you have any ethical concerns with this paper?

No

Have you any concerns about statistical analyses in this paper?

No

Recommendation?

Accept with minor revision (please list in comments)

Comments to the Author(s)

General Comments to the Author

The manuscript entitled 'Sprinting with prosthetic versus biological legs: an unfair advantage?', has the potential to be a high-quality article and an excellent reference for readers interested in the area. The manuscript is written to a high standard and offers unique insight to key parameters from the fastest BA on record. Whilst there are aspects of the manuscript that are excellent, there are key areas in which additional detail/ depth are required.

Research Design/ Methods.

Throughout the manuscript, there needs to be greater reflection/ acknowledgement of the threats to validity with some of the comparisons made across multiple studies with different methodological designs/ environments (lab vs. field). Whilst the reviewer understands the approach taken, the authors must consider the limitations of the current research design, but also provide clear recommendations for future experimental studies. In addition, the methods section should include more detail on how the NA data were obtained and treated.

Discussion of Results.

The results are presented in a logical order and provide clear comparison between the samples of athletes. However, in places there needs to be greater analysis of the results. Specifically, on page 16, lines 10-26 the explanation of curve running, and sprint endurance is comparatively weaker than the other sections of results. Given that the greatest benefit of the prosthetic is perceived to be in the final stages of the race (at higher speeds), the final 100 m split results could be explained in more detail, especially considering the findings for endurance metrics. If the current data cannot provide full explanation, what can future research endeavour to achieve?

Specific Comments:

Abstract, Line 13. Consider being more specific here toward the 'longer' sprint events.

Abstract, Line 18-23. The results for final split time showed a statistically faster time compared with NA, which is not currently considered in this statement.

Review form: Reviewer 3

Is the manuscript scientifically sound in its present form?

Yes

Are the interpretations and conclusions justified by the results?

No

Is the language acceptable?

Yes

Do you have any ethical concerns with this paper?

No

Have you any concerns about statistical analyses in this paper?

No

Recommendation?

Reject

Comments to the Author(s)

The paper is very informative and the authors clearly put a lot of effort in it. The manuscript is well written.

However, the paper fails to be able to answer the research question it wants to answer.

Other than previous work on this topic (e.g. Brüggemann 2008 et al) the authors include other important phases of a 400m race, for example the acc. phase and the curve running phase (even those they did a methodical error in curve sprinting, which leads to data hard to interpret).

However, the authors fail to provide data on the later part of the race. They point out the slower starting phase, some similarity during the top speed phase but leave out the most interesting phase of the later race, where non-amputee athletes get slower and athletes with bilateral amputation maintain their running velocity. The authors show better running economy by athletes with bilateral amputation compared to non-amputee athletes (contrary to their initial statement, that there was not a single parameter, bilateral amputee athletes had better values) but fail to provide biomechanics on the part of the race where this gets effective. This seems yet again to be a presentation of a one-sided point of view, which is sad because the data shown is interesting and a contribution to the field. However, in the current version this would lead to less comprehensive information and promote wrong conclusions by neglecting important factors. Given the declared conflict of interest I cannot recommend publication without biomechanical data on the second half of a 400m race. Decision makers might be guided in wrong directions by strong wording, which lacks full scientific support.

Detailed comments below:

Supplementary data:

No raw data, no scripts available. Only excel spread sheets with processed data.

Title: The terms fair and unfair are of rather philosophical nature. The manuscript does not and cannot answer this question in the current version.

Abstract:

Line 18-24: Too general and too strong statement. Also in your own manuscript contradict this statement.

Introduction P4: Line 39-53. Too long and should not be part of a biomechanical paper. The informed reader will know that. Are you trying to redirect the reader to a certain point?

Introduction P5: Line 11: no politics in a biomechanical paper, please.

Introduction P6: BA was not introduced yet.

Results P10: Line 29-31 Assuming a slow start I am impressed the athlete only was 6% slower in the curve compared to max sprinting. Within 20- 40 meter he most likely had not yet reach top speed. How did you ensure top speed was reached?

Results P10: Line 31-33 not correct. There are publications on that.

Results P10: Line 36-45 This statement cannot not be done with the methods applied. Velocity between 20-40 m from a standing start is simply to the right sector to determine max. curve sprinting velocity and compare it to max straight sprinting velocity.

Figure 4. How does other available data fit in here? E.g. from Cologne group? They published some curve sprinting data including for persons with bilateral transtibial amputation.

Results P12: Line 38-47: So there is a parameter which is better!!! At the beginning you stated very strongly there is non. Readers and decisions makers (who might only read abstracts) are getting guided in wrong directions.

Results P13: Line 3-8: So with other words: Athletes with bilateral amputation run more economic, and have less aerobic capacity. Thus, they can hold speed with less respiratory effort. Due to better running economy there is no need to training on respiratory capacity as much as non-amputees need to do. Appears to be an advantage.

Conclusion: Again strong statements, which are arguably by the manuscript itself, are possibly guided readers and decision makers in wrong directions.

Biomechanical data on the second part of the race, where non-amputee athlete begin to slow down are missing completely. Therefore, the point of view presented is only one-sided and neglects the effect of the respiratory differences on biomechanical parameters (e.g. GRF application, step length).

Decision letter (RSOS-211046.R0)

Dear Dr Beck

The Editors assigned to your paper RSOS-211046 "Sprinting with prosthetic versus biological legs: an unfair advantage?" have made a decision based on their reading of the paper and any comments received from reviewers.

Regrettably, in view of the reports received, the manuscript has been rejected in its current form. However, a new manuscript may be submitted which takes into consideration these comments.

We invite you to respond to the comments supplied below and prepare a resubmission of your manuscript. Below the referees' and Editors' comments (where applicable) we provide additional requirements. We provide guidance below to help you prepare your revision.

Please note that resubmitting your manuscript does not guarantee eventual acceptance, and we do not generally allow multiple rounds of revision and resubmission, so we urge you to make every effort to fully address all of the comments at this stage. If deemed necessary by the Editors, your manuscript will be sent back to one or more of the original reviewers for assessment. If the original reviewers are not available, we may invite new reviewers.

Please resubmit your revised manuscript and required files (see below) no later than 29-Mar-2022. Note: the ScholarOne system will 'lock' if resubmission is attempted on or after this deadline. If you do not think you will be able to meet this deadline, please contact the editorial office immediately.

Please note article processing charges apply to papers accepted for publication in Royal Society Open Science (<https://royalsocietypublishing.org/rsos/charges>). Charges will also apply to papers transferred to the journal from other Royal Society Publishing journals, as well as papers submitted as part of our collaboration with the Royal Society of Chemistry (<https://royalsocietypublishing.org/rsos/chemistry>). Fee waivers are available but must be requested when you submit your manuscript (<https://royalsocietypublishing.org/rsos/waivers>).

Thank you for submitting your manuscript to Royal Society Open Science and we look forward to receiving your resubmission. If you have any questions at all, please do not hesitate to get in touch.

on behalf of Dr Omid Kavehei (Associate Editor) and Kevin Padian (Subject Editor)
openscience@royalsociety.org

Editor comments:

Thanks for submitting your very interesting paper. I am not sure that it is really appropriate for our Organismal Biology section, and it probably belongs more in a clinical sports journal because there is not much biology in it; however, the editorial staff considered it appropriate and I hope you will find the reviews useful and appropriate as well.

This is a contentious topic on biomechanical, sporting, and apparently legal grounds, and I agree with the reviewer that your manuscript, to be suitable for RSOS, must be free of political and legal issues; thus, the "unfair" label should be dropped as well from your statements.

Your question is really whether BA runners have an artificial enhancement that statistically puts them faster than NA runners. This is difficult to answer on the basis of your data (and perhaps anyone's) because your BA sample size is so small. Ideally you would want data from single individuals who were runners before and after amputation, regardless of Olympic qualification, because you have no way to know whether BA athletes would have been competitive before

prosthesis. I agree with the reviewer that you should be forthright about such limitations of your study. You are clearly trying to make the argument that prosthesis does not “unfairly” enhance running performance, which seems to me to be a bias because the situation is complex.

For example, your Table 4 shows that a BA athlete performs less well up to 200m, and you use this to argue a lack of advantage. But performance substantially improves after that, to the point of surpassing all other NA athletes. So are we not talking about the length of the race as a critical factor? And at what point would curves and inclines, for example, make a difference? For these reasons I don't think that your final conclusion is incorrectly stated: they certainly do appear to have an advantage in the second half of a 400-m run, although your sample size is too small for any conclusion.

In my view a strictly biological paper would be interested in these kinds of questions rather than in whether BA athletes should or should not be disqualified from athletic competitions. I admit some discomfort in appearing to provide the imprimatur of the Royal Society on a thesis that has such obvious sociopolitical implications. If you can convert this manuscript into a strictly biological treatment, we would entertain a resubmission. Thanks for submitting and best wishes.

Reviewer comments to Author:

Reviewer: 1

Comments to the Author(s)

Dear authors,

This is a contemporary topic, and it will remain as such for years to come. Original information/data collected by established scientists and laboratories should be made available to the rest of the scientific community to study it further and debate it if necessary. Therefore, given the nature of the topic, my approach is to facilitate the dissemination of such research as it will enable us to synthesise a consensus on the use of artificial limbs in sport and make a positive difference to amputee and non-amputee athletes.

Whilst I am comfortable with your approach and the reasoning you are developing, I feel that that the key messages (abstract, conclusions and in other parts) are going beyond the evidence collected.

You have quantified for first time performance and physiological parameters from an elite amputee, and this is important for our knowledge and understanding but I suggest being cautious when trying to generalise the applicability of the findings. I would suggest stating that “the use of prosthetic legs during 400 m running races cannot be considered a priori advantageous” but I would refrain from more general statements referring to all amputee and non-amputee athletes and the application of your findings to other events. Apart from the small sample size the study is affected by a few limitations preventing us to draw robust and universal conclusions. For example, data have been collected in a different way than previous studies but also you are contrasting your athlete's data against only a small range of previous studies which measured similar variables in non-amputee athletes. There are a lot more studies in the literature with data from sprint starts, sprinting mechanics and performance which you have not included in your comparative past literature. On some occasions you are using review articles (e.g., Mero et al., 1992) rather than the actual original research to pull relevant data. There are two options here: you either do a full search of literature and enrich your sources (and redo your statistics) or you keep the existing sources but make a clear statement that this is indicative literature. This has an implication in your statistical analysis. I recognise that the approach to use 2SDs as a boundary to identify similarities/dissimilarities has been used before in a similar setting/article but given your sources do not include all key previous studies/data this method is rather underpowered here. I would just present the whole project as a case study and provide

comparisons only through absolute and relative differences. If you prefer to keep the 2SD metric too, please ensure that the limitations of this technique are discussed briefly.

Furthermore, I would consider an addition to the title: Sprinting with prosthetic versus biological legs: an unfair advantage? Interpretations from authentic data (or something similar).

Please consider the above and modify your introduction, core arguments, conclusions and abstract accordingly. A limitations section/paragraph should be included before drawing final conclusions.

Please see below more specific comments:

INTRODUCTION

P3, Line 6: replace “exceptional” with “male”. Their standing is demonstrated from information further down

P3, Line 32: change to “non-amputee Olympians”

P3, Line 32: replace “stifled” with “prevented” or similar

P3, Line 39: change from “the governing body of Olympic athletics” to “international governing body for the sport of athletics”

P4, Line 25: Change from “Alternatively, other scientists propose...” to “Alternatively, other scientists, including our research group, propose”

P4, Line 49: “Fastest athlete”? Need some more context, e.g., fastest on earth, fastest last year, or similar.

P4, Line 49: Add “following” between “the” and “400 m”

P5, line 7: Change “would have placed 7th at the 2016 Olympics” to “would have placed him 7th amongst the 2016 Olympic non-amputee finalists” or something similar.

METHODS

P19, Line 22: correct to “consent”

P20, Lines 8-13: What do you mean by “acceleration phase”? Is this the push-off phase? Please use a term more relevant to sprint start mechanics. Having a look at your 2014 sprint start paper (<http://dx.doi.org/10.1123/jab.2013-0113>) I could see different thresholds defined there. This is not a problem, but if there are different you may need a very brief justification, given you are citing this reference.

P22, Lines 8-10: Can you please clarify if these were three sprints per radius condition?

P22, Line 15: provide resolution information for the Casio camera. Also, report the zoom factor used as the camera was placed far away from the plane of action.

P23, Line 3: Specify make and model of treadmill

P23, Line 31: Specify make and model of gas analysis system

RESULTS

P5, Lines 43-45: By stating “No other athlete with prosthetic legs has had their lab-based 0 to 20 m running time published” do you imply that your test was lab-based? If this is the intention, then you would need to remove the term “lab-based” as these were not pure lab conditions. Perhaps you can say, “experimentally recorded” or something similar.

P6, Lines 3-6: Please remove “Altogether, athletes with prosthetic legs do not accelerate faster than non-amputee athletes at the beginning of a race”.

Also remove the following from Figure 1 caption: “Athletes with prosthetic legs accelerate slower at the beginning of a race than non-amputee athletes”.

The above statements are too universal and are not supported by your data. Even though it appears plausible you cannot categorically say this as you have not experimentally tested all amputees and non-amputee athletes.

P7, Table 1: The data from the Mero et al (1992) review paper for the vGRF refers to the maximal and not average force from those studies (check again table I from Mero et al 1992).

P7, Line 55: How did you ensure that your measurements were taken at 10 m/s? Also, are you sure that the reference number 19 applies here, and not number 17? I can't find any non-amputee data at 10m/s in Weyand et al, 2000.

P9, Lines 38-45: Please use this version: "Based on these data, the studied athletes with prosthetic legs do not have a relatively faster maximum curve running velocity than non-amputee athletes tested previously since their maximum curve-running velocity is ~1-3% slower than those of non-amputee athletes".

P10, Line 39: Start the sentence as: "The velocity of 5.0 m/s ...".

P11, Figure 5 caption: Remove "Non-amputee athletes have a faster running velocity at aerobic capacity than athletes with prosthetic legs".

P11, Line 52: replace "worse" with "lower"

P12, Line 45: use "the studied 400 m athletes..."

P14, Line 29: remove "statistically"

P15, Table 4: how did you obtain the split times for the Fastest BA? Was it from TV footage? Please provide relevant information in the methodology section.

P15, main paragraph: As suggested earlier, please refrain from using lab-based for track measurements with the radar gun and replace with "experimental" or similar.

P15, Lines 53 - P16, Line 10: I would add one more reason for the discrepancies and this is the fact that the velocities have been calculated via different methods (split times vs treadmill mechanics)

BL400 Data file: Starts Tab: Peak hGRF is not in BW as indicated.

Reviewer: 2

Comments to the Author(s)

General Comments to the Author

The manuscript entitled 'Sprinting with prosthetic versus biological legs: an unfair advantage?', has the potential to be a high-quality article and an excellent reference for readers interested in the area. The manuscript is written to a high standard and offers unique insight to key parameters from the fastest BA on record. Whilst there are aspects of the manuscript that are excellent, there are key areas in which additional detail/ depth are required.

Research Design/ Methods.

Throughout the manuscript, there needs to be greater reflection/ acknowledgement of the threats to validity with some of the comparisons made across multiple studies with different methodological designs/ environments (lab vs. field). Whilst the reviewer understands the approach taken, the authors must consider the limitations of the current research design, but also provide clear recommendations for future experimental studies. In addition, the methods section should include more detail on how the NA data were obtained and treated.

Discussion of Results.

The results are presented in a logical order and provide clear comparison between the samples of athletes. However, in places there needs to be greater analysis of the results. Specifically, on page 16, lines 10-26 the explanation of curve running, and sprint endurance is comparatively weaker than the other sections of results. Given that the greatest benefit of the prosthetic is perceived to be in the final stages of the race (at higher speeds), the final 100 m split results could be explained in more detail, especially considering the findings for endurance metrics. If the current data cannot provide full explanation, what can future research endeavour to achieve?

Specific Comments:

Abstract, Line 13. Consider being more specific here toward the 'longer' sprint events.

Abstract, Line 18-23. The results for final split time showed a statistically faster time compared with NA, which is not currently considered in this statement.

Reviewer: 3

Comments to the Author(s)

The paper is very informative and the authors clearly put a lot of effort in it. The manuscript is well written.

However, the paper fails to be able to answer the research question it wants to answer.

Other than previous work on this topic (e.g. Brüggemann 2008 et al) the authors include other important phases of a 400m race, for example the acc. phase and the curve running phase (even those they did a methodical error in curve sprinting, which leads to data hard to interpret).

However, the authors fail to provide data on the later part of the race. They point out the slower starting phase, some similarity during the top speed phase but leave out the most interesting phase of the later race, where non-amputee athletes get slower and athletes with bilateral amputation maintain their running velocity. The authors show better running economy by athletes with bilateral amputation compared to non-amputee athletes (contrary to their initial statement, that there was not a single parameter, bilateral amputee athletes had better values) but fail to provide biomechanics on the part of the race where this gets effective. This seems yet again to be a presentation of a one-sided point of view, which is sad because the data shown is interesting and a contribution to the field. However, in the current version this would lead to less comprehensive information and promote wrong conclusions by neglecting important factors. Given the declared conflict of interest I cannot recommend publication without biomechanical data on the second half of a 400m race. Decision makers might be guided in wrong directions by strong wording, which lacks full scientific support.

Detailed comments below:

Supplementary data:

No raw data, no scripts available. Only excel spread sheets with processed data.

Title: The terms fair and unfair are of rather philosophical nature. The manuscript does not and cannot answer this question in the current version.

Abstract:

Line 18-24: Too general and too strong statement. Also in your own manuscript contradict this statement.

Introduction P4: Line 39-53. Too long and should not be part of a biomechanical paper. The informed reader will know that. Are you trying to redirect the reader to a certain point?

Introduction P5: Line 11: no politics in a biomechanical paper, please.

Introduction P6: BA was not introduced yet.

Results P10: Line 29-31 Assuming a slow start I am impressed the athlete only was 6% slower in the curve compared to max sprinting. Within 20- 40 meter he most likely had not yet reach top speed. How did you ensure top speed was reached?

Results P10: Line 31-33 not correct. There are publications on that.

Results P10: Line 36-45 This statement cannot not be done with the methods applied. Velocity between 20-40 m from a standing start is simply to the right sector to determine max. curve sprinting velocity and compare it to max straight sprinting velocity.

Figure 4. How does other available data fit in here? E.g. from Cologne group? They published some curve sprinting data including for persons with bilateral transtibial amputation.

Results P12: Line 38-47: So there is a parameter which is better!!! At the beginning you stated very strongly there is non. Readers and decisions makers (who might only read abstracts) are getting guided in wrong directions.

Results P13: Line 3-8: So with other words: Athletes with bilateral amputation run more economic, and have less aerobic capacity. Thus, they can hold speed with less respiratory effort. Due to better running economy there is no need to training on respiratory capacity as much as non-amputees need to do. Appears to be an advantage.

Conclusion: Again strong statements, which are arguably by the manuscript itself, are possibly guided readers and decision makers in wrong directions.

Biomechanical data on the second part of the race, where non-amputee athlete begin to slow down are missing completely. Therefore, the point of view presented is only one-sided and neglects the effect of the respiratory differences on biomechanical parameters (e.g. GRF application, step length).

===PREPARING YOUR MANUSCRIPT===

If you have been asked to revise the written English in your submission as a condition of publication, you must do so, and you are expected to provide evidence that you have received language editing support. The journal would prefer that you use a professional language editing service and provide a certificate of editing, but a signed letter from a colleague who is a native speaker of English is acceptable. Note the journal has arranged a number of discounts for authors

using professional language editing services
(<https://royalsociety.org/journals/authors/benefits/language-editing/>).

===PREPARING YOUR REVISION IN SCHOLARONE===

<https://royalsociety.org/journals/authors/author-guidelines/#supplementary-material> to include a suitable title and informative caption. An example of appropriate titling and captioning may be found at https://figshare.com/articles/Table_S2_from_Is_there_a_trade-

off_between_peak_performance_and_performance_breadth_across_temperatures_for_aerobic_sc
ope_in_teleost_fishes_/3843624.

Author's Response to Decision Letter for (RSOS-211046.R0)

See Appendix A.

Decision letter (RSOS-211799.R0)

Dear Dr Beck,

I am pleased to inform you that your manuscript entitled "Sprinting with prosthetic versus biological legs: insight from experimental data" is now accepted for publication in Royal Society Open Science.

Please see the Royal Society Publishing guidance on how you may share your accepted author manuscript at <https://royalsociety.org/journals/ethics-policies/media-embargo/>. After publication, some additional ways to effectively promote your article can also be found here

<https://royalsociety.org/blog/2020/07/promoting-your-latest-paper-and-tracking-your-results/>.

on behalf of Dr Omid Kavehei (Associate Editor) and Kevin Padian (Subject Editor)
openscience@royalsociety.org

Appendix A

We thank each reviewer for their thoughtful and constructive comments. By addressing these comments, we believe that the quality of our manuscript has been improved. To highlight a few changes that we made to the manuscript, we now added a formal “Limitations & Future Research” section, more information regarding our testing protocol as well as those from the studies of non-amputee athletes, and additional information and interpretation regarding 400 m race splits. Please see our responses below to each reviewer comment (in underlined text).

Reviewer comments to Author:

Reviewer: 1

Comments to the Author(s)

Dear authors,

This is a contemporary topic, and it will remain as such for years to come. Original information/data collected by established scientists and laboratories should be made available to the rest of the scientific community to study it further and debate it if necessary. Therefore, given the nature of the topic, my approach is to facilitate the dissemination of such research as it will enable us to synthesise a consensus on the use of artificial limbs in sport and make a positive difference to amputee and non-amputee athletes.

- Thank you for your positive remarks and forward-thinking approach.

Whilst I am comfortable with your approach and the reasoning you are developing, I feel that that the key messages (abstract, conclusions and in other parts) are going beyond the evidence collected. You have quantified for first time performance and physiological parameters from an elite amputee, and this is important for our knowledge and understanding but I suggest being cautious when trying to generalise the applicability of the findings. I would suggest stating that “the use of prosthetic legs during 400 m running races cannot be considered a priory advantageous” but I would refrain from more general statements referring to all amputee and non-amputee athletes and the application of your findings to other events.

- We agree and have updated the manuscript throughout to be more specific when interpreting the data and implications. For specific text where we made changes to the interpretation of the results, see our replies to your following comments.

Regarding our conclusion sentences, we appreciate your suggestion, but feel that it is more conservative to keep “cannot be considered *unequivocally* advantageous” rather than “cannot be considered *a priory* advantageous”. People’s preconceived ideas can lead them to *a priory* consider prostheses advantageous. But based on all the relevant data, there remains some uncertainty whether there is an overall advantage or not – hence there is not unequivocally an advantage. Please see below for our updated conclusion sentences that are more specific to the data.

Lines 29-30: Therefore, based on these 400 m performance metrics, use of prosthetic legs during 400 m running races is not unequivocally advantageous compared to the use of biological legs.

Lines 347-350: Therefore, based on experimentally derived 400 m performance metrics, athletes with bilateral leg amputations using passive running-prostheses cannot be unequivocally considered to have an advantage over non-amputee athletes during 400 m competition.

Apart from the small sample size the study is affected by a few limitations preventing us to draw robust and universal conclusions. For example, data have been collected in a different way than previous studies but also you are contrasting your athlete's data against only a small range of previous studies which measured similar variables in non-amputee athletes. There are a lot more studies in the literature with data from sprint starts, sprinting mechanics and performance which you have not included in your comparative past literature. On some occasions you are using review articles (e.g., Mero et al., 1992) rather than the actual original research to pull relevant data. There are two options here: you either do a full search of literature and enrich your sources (and redo your statistics) or you keep the existing sources but make a clear statement that this is indicative literature. This has an implication in your statistical analysis. I recognise that the approach to use 2SDs as a boundary to identify similarities/dissimilarities has been used before in a similar setting/article but given your sources do not include all key previous studies/data this method is rather underpowered here. I would just present the whole project as a case study and provide comparisons only through absolute and relative differences. If you prefer to keep the 2SD metric too, please ensure that the limitations of this technique are discussed briefly.

- We acknowledge the different approaches for comparing the performance metrics of athletes with prosthetic versus biological legs. In our study, we prefer to keep the current indicative comparisons and assess whether there are differences between athlete cohorts using the range of the selected data and a 2 SD difference from the mean (consistent with Weyand et al. 2009 *J Appl Physiol*). We agree that there are potential limitations to this approach, and have added these limitations throughout the manuscript (see below).

Lines 335-338: Notably, the athlete comparisons were not exhaustive, were potentially statistically underpowered, and subtle differences between experiments may have influenced these comparisons (e.g., indoor versus outdoor track testing). We implore future studies to improve models of running performance and to use consistent protocols to compare data between studies.

Lines 514-517: Additionally, because we quantified 400 m performance metrics from the fastest BA using protocols that emulated those of previous studies, our statistical comparisons were selective, not comprehensive, and potentially underpowered.

Furthermore, I would consider an addition to the title: Sprinting with prosthetic versus biological legs: an unfair advantage? Interpretations from authentic data (or something similar).

- Combining your suggestion with those of the other reviewers and editor, we updated our title to the following: “Title: Sprinting with prosthetic versus biological legs: insight from experimental data”.

Please consider the above and modify your introduction, core arguments, conclusions and abstract accordingly. A limitations section/paragraph should be included before drawing final conclusions.

- We agree, and as mentioned above we modified the conclusions to address our results, and added a limitations section prior to the conclusions (lines 327-340).

Please see below more specific comments:

INTRODUCTION

P3, Line 6: replace “exceptional” with “male”. Their standing is demonstrated from information further down.

- We now specify ‘male’ in multiple places, including the indicated line number where we replaced ‘exceptional’ with ‘male’.

P3, Line 32: change to “non-amputee Olympians”

- We made the suggested change.

P3, Line 32: replace “stifled” with “prevented” or similar

- We changed “stifled” to “prevented”.

P3, Line 39: change from “the governing body of Olympic athletics” to “international governing body for the sport of athletics”

- We agree and changed the text as suggested.

P4, Line 25: Change from “Alternatively, other scientists propose...” to “Alternatively, other scientists, including our research group, propose”

- We updated the indicated sentence to read, Lines 65-67: “Alternatively, other scientists, including those from our research group, propose that...”

P4, Line 49: “Fastest athlete”? Need some more context, e.g., fastest on earth, fastest last year, or similar.

- We added more context to the sentence. Lines 74-76: To accomplish this goal, we measured the following 400 m performance metrics from the athlete who ran the fastest-ever 400 m time using prosthetic legs (fastest BA) following his competition season where he ran 400 m in 44.42 s.

P4, Line 49: Add “following” between “the” and “400 m”

- We made the suggested change.

P5, line 7: Change “would have placed 7th at the 2016 Olympics” to “would have placed him 7th amongst the 2016 Olympic non-amputee finalists” or something similar.

- Now that the 2021 Tokyo Olympics have taken place, we updated the indicated sentence to read: Lines 79-80: For context, a 44.42 s 400 m performance would have placed 6th at the 2021 Olympic Men's Finals.

METHODS

P19, Line 22: correct to “consent”

- Thank you for catching this important typo. We changed “consistent” to “consent”.

P20, Lines 8-13: What do you mean by “acceleration phase”? Is this the push-off phase? Please use a term more relevant to sprint start mechanics. Having a look at your 2014 sprint start paper (<http://dx.doi.org/10.1123/jab.2013-0113>) I could see different thresholds defined there. This is not a problem, but if there are different you may need a very brief justification, given you are citing this reference.

- We changed the term “acceleration phase” to “push-off phase”, and agree that this is more consistent with the cited literature (Lines 370-380).

Our experimental set-up did not allow us to mount the same “winged” plate used in Taboga et al. 2014, therefore the hands of the fastest BA in the “set” position were not completely on the front force plate. This prevented us from measuring any potential horizontal forces applied by the hands. We now specify this in the methods section.

Lines 379-381: We used a higher horizontal GRF threshold (20 N) for the beginning of the push-off phase compared to Taboga et al. (11) (0 N) because the fastest BA's hands were not completely placed on the front force plate in the “set” position.

P22, Lines 8-10: Can you please clarify if these were three sprints per radius condition?

- We updated the sentence as suggested.
Lines 429-433: The fastest BA performed three sprints per straightaway and curve-radius condition (two curve radii), with at least 8 minutes of rest between each trial, which is consistent with the protocol performed by Taboga et al. (Table 1) (13). Additionally, data from non-amputee athletes (32) (grey symbols in Fig. 4) involved three 60 m sprints with 8 minutes of rest between each trial on two separate days.

P22, Line 15: provide resolution information for the Casio camera. Also, report the zoom factor used as the camera was placed far away from the plane of action.

- We used a high-resolution setting on the Casio camera at 240 Hz and zoomed the camera so that the fastest BA filled the screen. We added representative photos and a video of the curve running protocol to enable others to emulate our protocol (see supplementary information “Curve Running Data Collection”).

P23, Line 3: Specify make and model of treadmill.

- We added the model of our treadmill “(Treadmetrix, Park City, UT)”.

P23, Line 31: Specify make and model of gas analysis system

- The make and model of the gas analysis system are given in the preceding paragraph. Lines 449-450: using expired gas analysis (ParvoMedics TrueOne 2400, Sandy, UT, USA). In the indicated paragraph, we describe the data analysis from the specified gas analysis system.

RESULTS

P5, Lines 43-45: By stating “No other athlete with prosthetic legs has had their lab-based 0 to 20 m running time published” do you imply that your test a was lab-test? If this is the intention, then you would need to remove the term “lab-based” as these were not pure lab conditions. Perhaps you can say, “experimentally recorded” or something similar.

- We updated the indicated sentence and modified other similar sentences to specify that these were *experimentally-derived* data (we no longer use “lab-based” terminology). Lines 95-96: No other athlete with bilateral prosthetic legs has had their 0 to 20 m running time published.

P6, Lines 3-6: Please remove “Altogether, athletes with prosthetic legs do not accelerate faster than non-amputee athletes at the beginning of a race”.

- We updated the conclusion sentence to be specific to the data and comparisons. Lines 101-103: Altogether, no experimentally-tested athlete with prosthetic legs has accelerated out of the starting blocks and run faster than elite non-amputee athletes over 20 m.

Also remove the following from Figure 1 caption: “Athletes with prosthetic legs accelerate slower at the beginning of a race than non-amputee athletes”. The above statements are too universal and are not supported by your data. Even though it appears plausible you cannot categorically say this as you have not experimentally tested all amputees and non-amputee athletes.

- We agree and deleted the indicated sentence in our figure caption (Line 105).

P7, Table 1: The data from the Mero et al (1992) review paper for the vGRF refers to the maximal and not average force from those studies (check again table I from Mero et al 1992).

- We now specify that Mero et al. 1992 report *net* vGRF values ($net\ vGRF = total\ vGRF - BW$) in table I. We added the following sentence in the caption of table 1 (see lines below):
Lines 126-128: Notably, Mero et al. (26) report net vGRF values (*i.e.*, $net\ vGRF = total\ vGRF - body\ weight\ (BW)$) and we report total vGRF to allow for comparisons with other reported values.

P7, Line 55: How did you ensure that your measurements were taken at 10 m/s?

- We added the answer to this question in the manuscript.
Lines 396-397: We calibrated the treadmill speeds prior to running trials using a Shimpo Tachometer (Electromatic Equip’t Co., Inc, Cedarhurst, NY).

Also, are you sure that the reference number 19 applies here, and not number 17? I can't find any non-amputee data at 10m/s in Weyand et al, 2000.

- Thank you for catching this error. Indeed, the proper reference is #17 (Weyand et al. 2009) not #19 (Weyand et al. 2010). We updated the manuscript accordingly.

P9, Lines 38-45: Please use this version: “Based on these data, the studied athletes with prosthetic legs do not have a relatively faster maximum curve running velocity than non-amputee athletes tested previously since their maximum curve-running velocity is ~1-3% slower than those of non-amputee athletes”.

- We updated our sentence to your suggested version:
Lines 182-184: Based on these data, the fastest BA does not have a relatively faster maximum curve running velocity than previously tested non-amputee athletes.

P10, Line 39: Start the sentence as: “The velocity of 5.0 m/s ...”.

- We made the indicated change.

P11, Figure 5 caption: Remove “Non-amputee athletes have a faster running velocity at aerobic capacity than athletes with prosthetic legs”.

- We deleted the indicated sentence.

P11, Line 52: replace “worse” with “lower”

- We replaced ‘worse’ with ‘lower’

P12, Line 45: use “the studied 400 m athletes...”

- We used “the studied 400 m athletes...”

P14, Line 29: remove “statistically”

- We deleted “statistically”

P15, Table 4: how did you obtain the split times for the Fastest BA? Was it from TV footage? Please provide relevant information in the methodology section.

- Thanks for the suggestion. We now added a “400 m Race Split” section in our methods to address your questions.

Lines 496-501: 400 m Race Splits. We quantified 100 m splits from the fastest BA and elite male non-amputee athletes during 400 m races. Specifically, we determined the fastest BA’s 100 m splits from a video recording of a 400 m race where he ran 44.42 s (<https://youtu.be/rqIvKYOlltw>) using digitizing software (Kinovea). From a published report (30), we indexed 100 m race splits of elite male non-amputee sprinters during the 400 m final of the IAAF World Championships in London, UK.

P15, main paragraph: As suggested earlier, please refrain from using lab-based for track measurements with the radar gun and replace with “experimental” or similar.

- Throughout the manuscript, we updated the text and no longer use the phrase ‘lab-based’.

P15, Lines 53 – P16, Line 10: I would add one more reason for the discrepancies and this is the fact that the velocities have been calculated via different methods (split times vs treadmill mechanics)

- We agree and added the suggested reason to the manuscript:
Lines 310-312: Fourth, differences in the precision of calculating an athlete's running velocity using different methods (e.g., calibrated treadmill velocity versus 25 Hz video recording of 100 m race splits) could have potentially affected the experimental- versus race-based comparisons.

BL400 Data file: Starts Tab: Peak hGRF is not in BW as indicated.

- Thank you for the catch, we corrected the header of the tab to indicate force in newtons (See supplementary file).

Reviewer: 2

Comments to the Author(s)

General Comments to the Author

The manuscript entitled ‘Sprinting with prosthetic versus biological legs: an unfair advantage?’, has the potential to be a high-quality article and an excellent reference for readers interested in the area. The manuscript is written to a high standard and offers unique insight to key parameters from the fastest BA on record. Whilst there are aspects of the manuscript that are excellent, there are key areas in which additional detail/ depth are required.

- Thank you for your positive remarks. We added more depth and detail to the manuscript, as indicated in the following responses to your comments.

Research Design/ Methods.

Throughout the manuscript, there needs to be greater reflection/ acknowledgement of the threats to validity with some of the comparisons made across multiple studies with different methodological designs/ environments (lab vs. field). Whilst the reviewer understands the approach taken, the authors must consider the limitations of the current research design, but also provide clear recommendations for future experimental studies. In addition, the methods section should include more detail on how the NA data were obtained and treated.

- We agree and have added a section that discusses potential limitations (including our comparisons with prior studies) and makes recommendations for future research – see the “Limitations and Future Directions” section. Moreover, in addition to citing the relevant non-amputee data comparisons throughout the manuscript, we added multiple sentences that further describe the protocol that we used as well as the comparison protocols throughout the methods section – including a new section on how we recorded athlete 400 m race split data. See below for specific updates.

Lines 330-342: Limitations and Future Directions.

We acknowledge that it is uncertain exactly how fast an athlete with prosthetic legs could run 400 m if they were a non-amputee athlete with biological legs using footwear (or vice-versa). Additionally, there is currently no published model that accurately predicts 400 m performance. Thus, in this study we compared 400 m performance metrics from athletes with bilateral leg amputations to those of non-amputee athletes who were tested experimentally using similar protocols. Notably, the athlete comparisons were not exhaustive, were potentially statistically underpowered, and subtle differences between experiments may have influenced these comparisons (e.g., indoor versus outdoor track testing). We implore future studies to improve models of running performance and to use consistent protocols to compare data between studies. Further, more research is warranted to determine why the 400 m race splits of athletes with bilateral leg amputations differ from those of non-amputee athletes. Such future research will help reveal how biomechanical and physiological factors affect running performance, which can be used to inform athletics rules and regulations.

Lines 362-363: During one of the testing days, the fastest BA warmed-up and then performed three maximum effort accelerations out of the starting blocks along a straightaway.

Lines 371-373: We compared data from the fastest BA to those of non-amputee athletes from a previous study who performed two maximum effort accelerations out of the starting blocks along a straightaway over distances of 0-10, 0-15, 0-20, 0-30, and 0-40 m (16).

Lines 405-406: The maximum velocity testing protocol was identical between the fastest BA, fastest athlete with a unilateral leg amputation (18), and fastest non-amputee athletes (29).

Lines 426-433: *Curve running.* On a separate day, the fastest BA warmed-up and then performed maximum effort 40 m sprints beginning from a standing start on a straightaway and counterclockwise curves that replicated lane 1 of a regulation 400 m outdoor track (radius; $r = 36.5$ m) and 200 m indoor track ($r = 17.2$ m) (Fig. 4) (13). The fastest BA performed three sprints per straightaway and curve-radius condition (two curve radii), with at least 8 minutes of rest between each trial, which is consistent with the protocol performed by Taboga et al. (Table 1) (13). Additionally, data from non-amputee athletes (32) (grey symbols in Fig. 4) involved three 60 m sprints with 8 minutes of rest between each trial on two separate days.

Lines 462-465: Our protocol was similar to the protocols from comparison studies, which assessed steady-state rates of oxygen uptake during 4-7 minute submaximal running trials (17, 34, 35), assessed blood lactate measures at the end of submaximal running trials (34, 35), and performed incremental aerobic capacity tests 10-20 minutes after the last submaximal running trial (34, 35).

Lines 476-482: On a separate day, the fastest BA performed six treadmill-running trials at velocities between his $\dot{V}O_{2peak}$ and maximum velocity (5.6, 6.2, 6.8, 7.8, 8.8, and 10.2 m/s). We randomized the trial order. Each trial was initiated by the fastest BA lowering himself from the handrails onto the moving treadmill belt. We measured the duration that the fastest BA could sustain each treadmill velocity using a stopwatch. Studies that involved non-amputee athletes and the 2nd fastest BA performed 2-6 sprint endurance trials per session, totaling 6-15 trials per participant (10, 17).

496-501: *400 m Race Splits.* We quantified 100 m splits from the fastest BA and elite male non-amputee athletes during 400 m races. Specifically, we determined the fastest BA's 100 m splits from a video recording of a 400 m race where he ran 44.42 s (<https://youtu.be/rqIvKYOlltw>) using digitizing software (Kinovea). From a published report (30), we indexed 100 m race splits of elite male non-amputee sprinters during the 400 m final of the IAAF World Championships in London, UK.

Lines 513-516: Additionally, because we quantified 400 m performance metrics from the fastest BA using protocols that emulated those of previous studies, our statistical comparisons were selective, not comprehensive, and potentially underpowered.

Discussion of Results.

The results are presented in a logical order and provide clear comparison between the samples of athletes. However, in places there needs to be greater analysis of the results. Specifically, on page 16, lines 10-26 the explanation of curve running, and sprint endurance is comparatively weaker than the other sections of results. Given that the greatest benefit of the prosthetic is perceived to be in the final stages of the race (at higher speeds), the final 100 m split results could be explained in more detail, especially considering the findings for endurance metrics. If the current data cannot provide full explanation, what can future research endeavour to achieve?

We agree with your comment and have added information regarding the curve running and sprint endurance results and the potential disconnect between these experimental results and race splits.

Lines 312-327: *Curve running.* Over the third 100 m split, both the fastest BA and elite non-amputee athletes slowed more than predicted (4.8% and 8.6%, respectively) based on their maximum curve running velocities. Athletes may have run relatively slower during their third race split than predicted due to racing a longer distance on the track versus the treadmill (20 vs. 100 m), in addition to altered pacing strategies and fatigue, which are not present in the experiments that measured maximum curve running velocity. Further complicating the experimental- versus race-based comparison, the fastest BA and non-amputee athletes (32) may have been accelerating throughout their straightaway and curve running experimental trials, whereas they were decelerating throughout the 3rd race split (200 m to 300 m) (Table 4). *Sprint Endurance.* Over the final 100 m split, the fastest BA was faster than elite non-amputee athletes despite having a similar sprint endurance profile (7, 10, 24). The difference between the sprint endurance profile versus the corresponding race splits may be related to typical variability in sprint endurance profiles, differences in race strategies, and/or environmental conditions. Alternatively, prosthetic legs may enable athletes to sustain relatively fast velocities for a longer duration than biological legs, despite nearly identical experimentally derived sprint endurance profiles (Fig. 6).

Lines 337-342: We implore future studies to improve models of running performance and to use consistent protocols to compare data between studies. Further, more research is warranted to determine why the 400 m race splits of athletes with bilateral leg amputations differ from those of non-amputee athletes. Such future research will help reveal how biomechanical and physiological factors affect running performance, which can be used to inform athletics rules and regulations.

Specific Comments:

Abstract, Line 13. Consider being more specific here toward the ‘longer’ sprint events.

Abstract, Line 18-23. The results for final split time showed a statistically faster time compared with NA, which is not currently considered in this statement.

- We agree and updated our abstract to specify ‘longer’ sprint events and to highlight final 400 m race splits.

Lines 17-19: Due to the world-class performances and relatively fast race finishes of these athletes, many people assume that running-prostheses provide users an unfair advantage over biologically-legged competitors during long sprint races.

Reviewer: 3

Comments to the Author(s)

The paper is very informative and the authors clearly put a lot of effort in it. The manuscript is well written.

- Thank you!

However, the paper fails to be able to answer the research question it wants to answer.

- We agree that the original research question was vague and not answered by our original submission. As such, we have revised the manuscript's research goal (see below) so that our analyses allow us to accomplish our goal.

Lines 72-88: Rather than reiterating theoretical arguments (7, 8), the goal of this study was to compare the data of athletes using bilateral prosthetic versus biological legs in experimental tests that relate to 400 m performance. To accomplish this goal, we measured the following 400 m performance metrics from the athlete who ran the fastest-ever 400 m time using prosthetic legs (fastest BA) following his competition season where he ran 400 m in 44.42 s: initial race acceleration (11, 16), maximum straightaway running velocity (3, 17-19), maximum curve running velocity (13, 20, 21), running velocity at aerobic capacity ($v\dot{V}O_{2peak}$) (17, 22, 23), and sprint endurance (10, 17, 24, 25). For context, a 44.42 s 400 m performance would have placed 6th at the 2021 Olympic Men's Finals. After testing the fastest BA's ability to perform each 400 m performance metric, we compared his results to those of other athletes with bilateral leg amputations using running-prostheses, including the 2nd fastest such 400 m athlete in history (Athlete: 2nd fastest BA) (9, 17). Subsequently, we compared the best performance metric value achieved across all athletes with prosthetic legs to the those across all non-amputee athletes. If any athlete with prosthetic legs exhibited a 400 m performance metric that was better than that observed by the best non-amputee athlete or over two standard deviations better than the average of elite non-amputee athletes (consistent with (17)), prosthetic legs likely confer a specific advantage in that metric compared to biological legs.

Other than previous work on this topic (e.g. Brüggemann 2008 et al) the authors include other important phases of a 400m race, for example the acc. phase and the curve running phase (even though they did a methodical error in curve sprinting, which leads to data hard to interpret).

- In a subsequent comment, we provide more detail regarding the curve sprinting data. Briefly, the curve running protocol performed by the fastest BA was the same as the protocol of Taboga et al. *J Exp Biol* 2016, and nearly the same as the protocol used for non-amputees in Churchill et al. *Sports Biomech* 2015. We compared the maximum curve running velocity of the fastest bilateral athlete to that of non-amputees in Churchill et al. as well as to an established model from Greene 1985 *J Biomech Eng* based on non-amputee athletes and confirmed with experimental data. It is possible that the fastest BA

(in this study and Taboga et al.) and non-amputee athletes (from Churchill) were still accelerating to reach their maximum velocity throughout the entire trial. We now acknowledge this potential limitation in the manuscript (See below). Yet, because we assessed both straightaway and curve running velocities on a track using the same protocol (e.g., average velocity at 20-40 m from the start), we can compare curve running velocity to straightaway velocity (see below).

Lines 318-321: Further complicating the experimental- versus race-based comparison, the fastest BA and non-amputee athletes (32) may have been accelerating throughout their straightaway and curve running experimental trials, whereas they were decelerating throughout the 3rd race split (200 m to 300 m) (Table 4).

Lines 429-433: The fastest BA performed three sprints per straightaway and curve-radius condition (two curve radii), with at least 8 minutes of rest between each trial, which is consistent with the protocol performed by Taboga et al. (Table 1) (13). Additionally, data from non-amputee athletes (32) (grey symbols in Fig. 4) involved three 60 m sprints with 8 minutes of rest between each trial on two separate days.

However, the authors fail to provide data on the later part of the race. They point out the slower starting phase, some similarity during the top speed phase but leave out the most interesting phase of the later race, where non-amputee athletes get slower and athletes with bilateral amputation maintain their running velocity. The authors show better running economy by athletes with bilateral amputation compared to non-amputee athletes (contrary to their initial statement, that there was not a single parameter, bilateral amputee athletes had better values) but fail to provide biomechanics on the part of the race where this gets effective.

- We assessed 400 m performance metrics that are collectively relevant to each race phase – from the beginning to the end of a race. Regarding the ability of an athlete to sustain a relatively fast running velocity for a longer duration during a 400 m race, we assessed two directly relevant performance metrics – an athlete’s velocity at $\dot{V}O_{2peak}$ ($v\dot{V}O_{2peak}$) and their sprint endurance profile – and found no difference between athletes with bilateral leg amputations and non-amputee athletes. We have added information regarding the $v\dot{V}O_{2peak}$ and sprint endurance results and the potential disconnect between these experimental results and race splits.

As stated in our manuscript, Lines 195-198: During a 400 m race, athletes expend metabolic energy via both anaerobic and aerobic metabolism (33). If other performance metrics are equal, the athlete who has a faster velocity at aerobic capacity ($v\dot{V}O_{2peak}$) will outperform others in a 400 m race (10, 24).

Thus, even if athletes with bilateral leg amputations are more economical runners than non-amputee sprinters, better economy does not necessarily constitute an overall advantage for a 400 m race – we also state this in our manuscript.

Lines 211-212: Because $v\dot{V}O_{2peak}$ depends on running economy and aerobic capacity ($\dot{V}O_{2peak}$), we also compared these parameters between athletes with and without bilateral leg amputations.

Since the tested athletes with bilateral leg amputations have a lower $\dot{V}O_{2peak}$ than non-amputee athletes, their fastest aerobic running velocity ($v\dot{V}O_{2peak}$) is slower than non-amputee distance runners and nearly identical to non-amputee sprinters. We state this in our manuscript, Lines 220-222: Thus, despite being relatively economical runners (15), the lower $\dot{V}O_{2peak}$ of the measured athletes with prosthetic legs contribute to a $v\dot{V}O_{2peak}$ that is not faster than that of non-amputee 400 m athletes and distance runners.

In other words, based on the available data, athletes with bilateral leg amputations cannot sustain a faster running velocity than non-amputees using primarily aerobic metabolism. Further, non-amputee athletes have been reported to achieve a lower metabolic cost of transport than the fastest BA (Lucia et al. 2008 *Br J Sports Med*).

Additionally, no athlete with bilateral leg amputations has been experimentally shown to be able to sustain a relatively fast running velocity for a longer duration than non-amputees (e.g., 90% of their maximum velocity for 100 seconds). Therefore, our results are consistent with previous findings (e.g., Weyand et al. 2009 *J Appl Physiol*) stating that athletes with prosthetic legs do not have superior sprinting endurance than non-amputee athletes. In our manuscript, we point out differences in 400 m race splits between athlete cohorts, and detail potential discrepancies between the experimental tests and actual race outcomes. We also acknowledge that experimentally-derived sprint endurance protocols (that have been experimentally verified with all-out track performances (Bundle et al. 2003)) cannot rule out a potentially faster race finish of athletes with prosthetic versus biological legs.

Lines 321-327: *Sprint Endurance*. Over the final 100 m split, the fastest BA was faster than elite non-amputee athletes despite having a similar sprint endurance profile (7, 10, 24). The difference between the sprint endurance profile versus the corresponding race splits may be related to typical variability in sprint endurance profiles, differences in race strategies, and/or environmental conditions. Alternatively, prosthetic legs may enable athletes to sustain relatively fast velocities for a longer duration than biological legs, despite nearly identical experimentally-derived sprint endurance profiles (Fig. 6).

Lines 339-342: More research is warranted to determine why the 400 m race splits of athletes with bilateral leg amputations differ from those of non-amputee athletes. Such future research will help reveal how biomechanical and physiological factors affect running performance, which can be used to inform athletics rules and regulations.

This seems yet again to be a presentation of a one-sided point of view, which is sad because the data shown is interesting and a contribution to the field. However, in the current version this would lead to less comprehensive information and promote wrong conclusions by neglecting important factors. Given the declared conflict of interest I cannot recommend publication

without biomechanical data on the second half of a 400m race. Decision makers might be guided in wrong directions by strong wording, which lacks full scientific support.

- We appreciate the sincerity of your comment and believe that we have addressed your concerns regarding any apparent “view” in the manuscript. We have revised the manuscript to be clearer about our goals, methodology, and comparisons with previous studies. We replicated established sprinting protocols of peer-reviewed published manuscripts to compare data of the fastest BA, 2nd fastest BA, and other athletes with prosthetic legs to non-amputee athletes. These comparisons of 400 m performance metrics are similar to those done in previous research (Weyand et al. 2009), which compared top speed, sprint endurance protocols, and $v\dot{V}O_{2\max}$ of the 2nd fastest athlete with prosthetic legs to different non-amputee athlete cohorts. In addition to these comparisons, we compared two more performance metrics and provided a larger sample size and faster cohort of athletes with leg amputations, which increased the odds of athletes with prosthetic legs outperforming non-amputees.

As mentioned earlier, we clarified our research goal to assess 400 m performance metrics of the athlete cohorts. We also updated the conclusions to be more specific to these tests. As outlined in the competing interests statement, two of the authors were provided with funds to cover travel and accommodations to collect data presented in this study. Notably, none of the authors has received any financial compensation throughout the entirety of this study and all relevant court cases – all authors provided their time *pro bono*.

Lines 532-537: *Competing interests*. The fastest BA provided funds to cover travel to and accommodations at the investigation site for two of the manuscript’s authors (ONB and PT). All authors provided their time *pro bono* throughout the entirety of this study, as well as during all legal cases. All authors were involved in the following legal cases: Court of Arbitration in Sport, Leeper vs. World Athletics, 2020 & 2021.

We have acknowledged our potential competing interests, detailed the potential limitations, tailored the text to be more specific to our data, and described how future research can further inform this research topic.

Detailed comments below:

Supplementary data:

No raw data, no scripts available. Only excel spread sheets with processed data.

- Thank you for the suggestion to upload raw data. We agree that the data from the fastest athlete with bilateral legs are important to provide so that readers can perform their own analyses. Thus, in addition to our summary excel spreadsheet, which presents analyzed data from the fastest BA during each 400 m performance metric (SummaryData_FastestBA.xlsx), we have uploaded a video, photos, and raw data files. Namely, we uploaded a representative curve running trial video (Curve Running Trial) and two curve running photos to demonstrate our video quality as well as how we

recorded maximum velocity for running on a curve (Curve Radius 36.5 m Start & Finish). Additionally, we added a spreadsheet of the raw running economy and aerobic capacity data from our indirect calorimetry system (MetabolicData_FastestBA.xlsx). We also uploaded the vertical ground reaction force data from the force-measuring treadmill during both maximum running velocity sessions (3 to 11.4 m/s; TreadmillGRFs_FastestBA.xlsx). We uploaded raw radar gun and force plate data from the starting block acceleration trials (Starts_FastestBA.xlsx). We would also like to highlight the figures and tables within the manuscript, some of which include raw data (e.g., Table 3 shows running duration per velocity). Furthermore, our methods are detailed such that readers can replicate the analyses using the raw data. Some of the measurements were simply made using a hand-help stopwatch (e.g., sprint-endurance), or averaging raw values (e.g., oxygen uptake during running), and we used reported thresholds and standard filtering to report ground reaction force data (e.g., treadmill biomechanics data).

Title: The terms fair and unfair are of rather philosophical nature. The manuscript does not and cannot answer this question in the current version.

- We agree and following the suggestions of the reviewers and editor, changed the title to, “Sprinting with prosthetic versus biological legs: insight from experimental data”.

Abstract:

Line 18-24: To general and to strong statement. Also in your own manuscript contradict this statement.

- We respectfully disagree. In our manuscript, we define the five 400 m performance metrics, all of which have been previously related to sprinting performance (see below). Supported by the results, no athlete with bilateral leg amputations, including the fastest such athletes, have exhibited a better initial race acceleration, maximum straight away running velocity, maximum curve running velocity, running velocity at aerobic capacity ($\dot{V}O_{2peak}$), or sprint endurance profile than that achieved by elite non-amputees. Thus, the results support the conclusions that athletes with prosthetic legs cannot unequivocally be considered to have an advantage versus non-amputees in a 400 m race.

Lines 72-79: Rather than reiterating theoretical arguments (7, 8), the goal of this study was to compare the data of athletes using bilateral prosthetic versus biological legs in experimental tests that relate to 400 m performance. To accomplish this goal, we measured the following 400 m performance metrics from the athlete who ran the fastest-ever 400 m time using prosthetic legs (fastest BA) following his competition season where he ran 400 m in 44.42 s: initial race acceleration (11, 16), maximum straightaway running velocity (3, 17-19), maximum curve running velocity (13, 20, 21), running velocity at aerobic capacity ($\dot{V}O_{2peak}$) (17, 22, 23), and sprint endurance (10, 17, 24, 25).

Lines 21-24: However, here we show that no athlete with bilateral leg amputations using running-prostheses, including the fastest such athlete, exhibits a single 400 m running performance metric that is superior to those achieved by non-amputee athletes.

Introduction P4: Line 39-53. Too long and should not be part of a biomechanical paper.

The informed reader will know that. Are you trying to redirect the reader to a certain point?

- In the introduction, we aimed to provide context, justification, and background information that is relevant to our research. We have revised the introduction to more clearly focus on our research goal.

Introduction P5: Line 11: no politics in a biomechanical paper, please.

- The indicated topic sentence was meant to transition from the previous paragraph regarding current athletics policies, and the lack of scientific consensus regarding whether biological or prosthetic legs are better for running 400 m (see lines below). We have revised the sentence.
Lines 59-61: Athletics regulations regarding the use of running-prostheses are hindered by the lack of scientific consensus regarding the *net effect* of running with prosthetic versus biological legs (7, 8).

Introduction P6: BA was not introduced yet.

- Thank you. We have revised these sentences accordingly.
Lines 74-76: To accomplish this goal, we measured the following 400 m performance metrics from the athlete who ran the fastest-ever 400 m time using prosthetic legs (fastest BA) following his competition season where he ran 400 m in 44.42 s...

Results P10: Line 29-31 Assuming a slow start I am impressed the athlete only was 6% slower in the curve compared to max sprinting. Within 20- 40 meter he most likely had not yet reach top speed. How did you ensure top speed was reached?

- Thank you. We have added more information about the maximum effort straightway and curve-running trials. We recorded the maximum straightaway running velocity and the maximum curve-running velocity on a track surface, and calculated the average velocity between 20-40 m. We have updated our methods section to clarify this information (See below).
We agree that the fastest BA may have still been accelerating from 20-40 m during the straightaway and curve-running trials. We now acknowledge that there is a potential limitation in that athletes may have been accelerating when we measured maximum velocity on the straightaway and on the curve (see below).

Lines 426-433: On a separate day, the fastest BA warmed-up and then performed maximum effort 40 m sprints beginning from a standing start on a straightaway and counterclockwise curves that replicated lane 1 of a regulation 400 m outdoor track (radius; $r = 36.5$ m) and 200 m indoor track ($r = 17.2$ m) (Fig. 4) (13). The fastest BA performed three sprints per straightaway and curve-radius condition (two curve radii), with at least 8 minutes of rest between each trial, which is consistent with the protocol performed by Taboga et al. (Table 1) (13). Additionally, data from non-amputee athletes (32) (grey symbols in Fig. 4) involved three 60 m sprints with 8 minutes of rest between each trial on two separate days.

Lines 315-321: Athletes may have run relatively slower during their third race split than predicted due to racing a longer distance on the track versus the treadmill (20 vs. 100 m), in addition to altered pacing strategies and fatigue, which are not present in the experiments that measured maximum curve running velocity. Further complicating the experimental- versus race-based comparison, the fastest BA and non-amputee athletes (32) may have been accelerating throughout their straightaway and curve running experimental trials, whereas they were decelerating throughout the 3rd race split (200 m to 300 m) (Table 4).

Results P10: Line 31-33 not correct. There are publications on that.

- We are not aware of any publications that experimentally compared maximum curve running velocity from an athlete with bilateral leg amputations to their maximum straightaway running velocity – aside from the present study. To our knowledge, no other study has published experimental data showing how much athletes with bilateral leg amputations slow on a curve versus straightaway. Funken et al. 2016 have a conference proceeding where they reported curve running velocity of an athlete with bilateral transtibial amputations, however straightaway velocity was not reported. Further, multiple studies have studied the curve-running biomechanics of athletes with *unilateral* amputations (Taboga et al. *J Exp Biol* 2016; Funken et al. *Sports Biomech* 2017; Li et al. *J Sports Sci* 2018).

Results P10: Line 36-45 This statement cannot not be done with the methods applied. Velocity between 20-40 m from a standing start is simply to the right sector to determine max. curve sprinting velocity and compare it to max straight sprinting velocity.

- As detailed in the previous comment, the maximum curve-running velocity for lane 1 of a regulation 400 m track curve for the fastest athlete with bilateral leg amputations is 6% slower than his comparable maximum straightaway velocity.

Figure 4. How does other available data fit in here? E.g. from Cologne group? They published some curve sprinting data including for persons with bilateral transtibial amputation.

- As mentioned above, we are not aware of any other published peer-reviewed articles that present experimental curve running and straightaway maximum velocities of athletes with bilateral transtibial amputations – including from the Cologne group. If such data do exist, we are happy to incorporate these data into our manuscript.

Results P12: Line 38-47: So there is a parameter which is better!!! At the beginning you stated very strongly there is non. Readers and decisions makers (who might only read abstracts) are getting guided in wrong directions.

- While athletes with bilateral leg amputations may be more economical runners than non-amputee sprinters, that does not by itself constitute a better 400 m performance metric. Rather, the athlete with a better 400 m performance metric has the ability to run faster at their $\dot{V}O_{2\text{peak}}$ ($v\dot{V}O_{2\text{peak}}$) (consistent with Weyand et al. 2009; Daniels *MSSE* 1985; Joyner

J Appl Physiol 1991). And since athletes with bilateral leg amputations had a lower $\dot{V}O_{2peak}$ than non-amputee athletes, their $v\dot{V}O_{2peak}$ was not superior to those of non-amputee sprinters. In other words, based on the available data, athletes with bilateral leg amputations cannot sustain a faster running velocity than non-amputee athletes using primarily aerobic metabolism. That is because the *relative* aerobic intensity at a given running velocity (e.g., % $\dot{V}O_{2peak}$) of athletes with bilateral leg amputations is not better than non-amputee athletes. Furthermore, non-amputee athletes have been reported to achieve the lowest *absolute* metabolic cost of transport (Lucia et al. 2008 *Br J Sports Med*), not athletes with bilateral leg amputations. See below for description in manuscript.

Lines 211-212: Because $v\dot{V}O_{2peak}$ depends on running economy and aerobic capacity ($\dot{V}O_{2peak}$), we also compared these parameters between athletes with and without bilateral leg amputations.

Lines 220-222: Thus, despite being relatively economical runners (15), the lower $\dot{V}O_{2peak}$ of the measured athletes with prosthetic legs contribute to a $v\dot{V}O_{2peak}$ that is not faster than that of non-amputee 400 m athletes and distance runners.

Results P13: Line 3-8: So with other words: Athletes with bilateral amputation run more economic, and have less aerobic capacity. Thus, they can hold speed with less respiratory effort. Due to better running economy there is no need to training on respiratory capacity as much as non-amputees need to do. Appears to be an advantage.

With all due respect, these suggestions are not supported by scientific data and do not address 400 m performance. The most economical athletes in the literature are elite non-amputee distance runners (Lucia et al. 2008 *Br J Sports Med*), not athletes with leg amputations (Beck & Grabowski 2019 *Exerc Sport Sci Rev*). Also, if the use of prosthetic legs enables an athlete to run more economically than the use of biological legs, they likely still need to “train as much (or more) than non-amputees”. That is because running velocity over 400 m depends on both the submaximal and maximal rate of oxygen uptake ($\dot{V}O_{2submax}$ and $\dot{V}O_{2max}$), as stated in our manuscript (e.g., Joyner 1991 *J Appl Physiol*). Because athletes with prosthetic legs do not have maximal rates of oxygen uptake that are as high as non-amputee athletes, their *relative* aerobic intensity is similar or worse than that of non-amputees at a given running velocity. Running velocity at $\dot{V}O_{2max}$ and at velocities slower than $v\dot{V}O_{2max}$ depend on *relative* aerobic intensity, not absolute aerobic intensity. In other words, athletes with leg amputations run at a similar or higher percentage of their $\dot{V}O_{2peak}$ than non-amputees at a given velocity. Simply, the *relative* respiratory effort of athletes with prosthetic legs is similar or higher than that of non-amputee athletes – thus, athletes with prosthetic legs do not have a better $v\dot{V}O_{2max}$ or run faster at a given relative effort than non-amputee athletes based on this 400 m performance metric. Non-amputee athletes run as fast or faster than athletes with prosthetic legs at $\dot{V}O_{2max}$ (Fig. 5; dashed line indicates $v\dot{V}O_{2max}$).

Figure 5. Submaximal and maximal rates of oxygen uptake ($\dot{V}O_2$) versus running velocity for non-amputee 400 m athletes (NA 400 m - silver circles) (17), non-amputee distance runners (NA Dist Run - black circles) (34), as well as the fastest (Fastest BA - red diamonds) and second fastest (2nd fastest BA - blue diamonds) 400 m athletes with bilateral leg amputations using running-prostheses. The greatest $\dot{V}O_2$ value ($\dot{V}O_{2peak}$) for each athlete and cohort is indicated by a square around the symbol. The vertical dashed lines indicate the velocity at $\dot{V}O_{2peak}$. Error bars are SD when applicable.

Conclusion: Again strong statements, which are arguably by the manuscript itself, are possibly guided readers and decision makers in wrong directions.

- Following the reviewers' suggestions, we have added a "Limitations and Future Directions" section to our manuscript and updated our interpretations and conclusions to be more specific to our data.

Biomechanical data on the second part of the race, where non-amputee athlete begin to slow down are missing completely. Therefore, the point of view presented is only one-sided and neglects the effect of the respiratory differences on biomechanical parameters (e.g. GRF application, step length).

- We provide biomechanical data that are indicative of the second part of the 400 m race. We present and compare 400 m performance metrics between athletes with prosthetic versus biological legs - including data from multiple Olympic-caliber 400 m athletes with prosthetic legs. These experimentally derived 400 m performance metrics include biomechanical data and affect the entire 400 m race, as detailed above. Our methods are consistent with previous publications that compare the sprinting abilities of athletes with and without amputations (e.g., Weyand et al. 2009 *J Appl Physiol*, Rabita et al. 2015 *Scand J Med Sci Sports*, Taboga et al. 2014 *J Exp Biol*).

We provide biomechanical data, including relevant stride kinematics and ground reaction forces, across a wide range of velocities (3 m/s to maximum velocity (11.4 m/s)) from the fastest and second fastest BA and compare them to those from non-amputee 400 m athletes. Moreover, we found no differences in these stride kinematics and ground reaction forces between the fastest BA and non-amputee 400 m athletes at a given velocity (Fig. 2 & 3).

We also show that the fastest and second fastest BA have the same sprint endurance profile as non-amputee athletes (Fig. 6). Further, we report 400 m race splits and acknowledge that the fastest athlete with bilateral amputations runs the same 400 m time as elite non-amputee athletes using slower race starts and faster race finishes. We discussed the differences between experimentally- and race-derived data in our manuscript (see below).

In addition to the sprint endurance profiles, and based on all available data, athletes with prosthetic legs run with relatively similar or greater relative aerobic/respiratory intensities as non-amputee athletes.

Lines 323-327: The difference between the sprint endurance profile versus the corresponding race splits may be related to typical variability in sprint endurance profiles, differences in race strategies, and/or environmental conditions. Alternatively, prosthetic legs may enable athletes to sustain relatively fast velocities for a longer duration than biological legs, despite nearly identical experimentally-derived sprint endurance profiles (Fig. 6).

Lines 339-340: More research is warranted to determine why the 400 m race splits of athletes with bilateral leg amputations differ from those of non-amputee athletes.